# ON THE GENERALIZATION OF WASSERSTEIN ROBUST FEDERATED LEARNING

## ABSTRACT

In Federated Learning (FL), participating clients typically possess non-i.i.d. data, posing a significant challenge to generalization to unseen distributions. To address this, we propose a Wasserstein distributionally robust optimization scheme called WAFL. Leveraging its duality, we frame WAFL as an empirical surrogate risk minimization problem, and solve it using a novel local SGD-based algorithm with convergence guarantees. We show that the robustness of WAFL is more general than related approaches, and the generalization bound is robust to all adversarial distributions inside the Wasserstein ball (ambiguity set). Since the center location and radius of the Wasserstein ball can be suitably modified, WAFL shows its applicability not only in robustness but also in domain adaptation. Through empirical evaluation, we demonstrate that WAFL generalizes better than the vanilla FedAvg in non-i.i.d. settings, and is more robust than other related methods in distribution shift settings. Further, using benchmark datasets we show that WAFL is capable of generalizing to unseen target domains.

## 1 INTRODUCTION

Federated Learning (FL) (Konečný et al., 2016; McMahan et al., 2017) has emerged as a cutting-edge technique in distributed and privacy-preserving machine learning. The nature of highly non-independent and identically distributed (non-i.i.d.) data in clients' devices poses an important challenge to FL commonly called *statistical heterogeneity*. The global model trained on this data using the de facto FedAvg algorithm (McMahan et al., 2017) has been shown to generalize poorly to individual clients' data, and further to *unseen distributions* on new clients as they enter the network.

Several solutions to data heterogeneity have been proposed. Personalized FL (Fallah et al., 2020; Deng et al., 2020a; Dinh et al., 2021; Li et al., 2021) and multi-task FL (Smith et al., 2018) are *client-adaptive* approaches, where a personalized model is adapted to each client. From another perspective, distributionally robust FL trains a model using a worst-case objective over an *ambiguity set* (Mohri et al., 2019; Du et al., 2020; Reisizadeh et al., 2020; Deng et al., 2020b). This approach is *client-uniform* because a single global model is judiciously learned to deliver uniformly good performance not only for all training clients but also for new/unseen clients with unknown data distributions. It is specifically useful when test distributions drift away from the training distributions.

A natural question when designing distributionally robust FL frameworks is generalization: how can minimizing the training error also bound the test error (*generalization bounds*)? In FL, Mohri et al. (2019) proposed agnostic FL where a model is designed to be robust against any ambiguity set as a convex combination of the clients' distributions. Reisizadeh et al. (2020) applied the general affine covariate shift – used in the standard adversarial robust training – into FL training. In characterizing the generalization bounds, while Mohri et al. (2019) relied on the standard Rademacher complexity, Reisizadeh et al. (2020) use the margin-based technique developed by Bartlett et al. (2017).

In this work, we take a different approach called WAsserstein distributionally robust FL (WAFL for short). The ambiguity set in WAFL is a Wasserstein ball of all adversarial distributions in close proximity to the nominal data distribution at the center. Our main contributions are:

- We propose a distributionally robust optimization problem to address statistical heterogeneity in FL. By controlling the center and radius of the Wasserstein ball, we show that WAFL is robust to a wider range of adversarial distributions than is agnostic or adversarial FL.

- To make WAFL more amenable to distributed optimization, we transform the original problem into a minimization of the empirical surrogate risk. We propose a local SGD-based algorithm to solve this surrogate problem. With additional Lipschitz smoothness conditions, standard techniques can be applied to find the convergence rate for the proposed algorithm.

- We show how WAFL's output can reduce the test error by bounding its excess risk. We call this the *robust generalization bound* as it is applicable to all adversarial distributions inside the Wasserstein ball. By scaling the Wasserstein radius based on local data sizes, we show this bound is applicable to the true (unknown) data distribution among all clients.

- We show WAFL's extra flexibility in controlling the location of the Wasserstein ball by adjusting the nominal distribution. This enables applications such as multi-source domain adaptation and robustness to all clients' unknown distributions with minimal Wasserstein radius.

- Experimentally, we discuss how to control this radius for a given center location by fine-tuning a robust hyperparameter. We show that WAFL generalizes better than FedAvg in non-i.i.d. settings and further outperforms existing robust methods in distribution shift settings. We finally explore WAFL's capability in transferring knowledge from multi-source domains to related target domains with much less data and/or without labels.

## 2    RELATED WORK

**Federated Learning** was introduced in response to three challenges of machine learning at scale: massive data quantities at the edge, communication-critical networks of participating devices, and privacy-preserving learning without central data storage (Konečný et al., 2016; McMahan et al., 2017). The de facto federated optimization algorithm – FedAvg (McMahan et al., 2017) – is based on local stochastic gradient descent (SGD) and averaging and is often considered a baseline in FL.

Most challenges of FL are categorized into *systems heterogeneity* and *statistical heterogeneity*. The former focuses on communication problems such as connection loss and bandwidth minimization. This motivated some prior works to design more communication-efficient methods (Konečný et al., 2016; 2017; Suresh et al., 2017). On the other hand, statitical heterogeneity is concerned with clients' non-i.i.d. data, which is the main cause behind aggregating very different models leading to one which does not perform well on any data distribution. To address this, many ideas have been introduced. Li et al. (2020) provided much theoretical analysis of FL non-i.i.d. settings. Zhao et al. (2018) proposed an FL framework which globally shares a small subset of data among clients to train the model with non-i.i.d. data. Smith et al. (2018) introduced a multi-task FL framework in which each client individually learns its own data pattern while borrowing information from other clients. Mansour et al. (2020) proposed three approaches to adapt the FL model to enable personalization, in reponse to distribution shift. Several personalized FL models have also been developed, including Fallah et al. (2020); Deng et al. (2020a); Dinh et al. (2021); Li et al. (2021).

**Wasserstein Distributionally Robust Optimization (WDRO)** aims to lean a robust model against adversarially manipulated data. The unknown data distribution is assumed to lie within a Wasserstein ball centered around the empirical distribution (Kuhn et al., 2019). WDRO has received attention as a promising tool for training parametric models, both in centralized and federated learning settings.

In centralized learning, many studies have proposed solutions based on WDRO problems for certain machine learning tasks (Shafieezadeh Abadeh et al., 2015; Gao & Kleywegt, 2016; Esfahani & Kuhn, 2017; Chen & Paschalidis, 2018; Sinha et al., 2020; Blanchet et al., 2019; Shafieezadeh-Abadeh et al., 2019; Gao et al., 2020). For instance, Shafieezadeh Abadeh et al. (2015) considered a robust logistic regression model under the assumption that the probability distributions lie in a Wasserstein ball. Chen & Paschalidis (2018); Blanchet et al. (2019); Gao et al. (2020) leveraged WDRO to recover regularization formulations in classification and regression. Gao & Kleywegt (2016) proposed a minimizer based on a tractable approximation of the local worst-case risk. Esfahani & Kuhn (2017) used WDRO to formulate the search for the largest perturbation range as an optimization problem and solve its dual problem. Sinha et al. (2020) introduced a robustness certificate based on a Lagrangian relaxation of the loss function which provably robust against adversarial input distributions within a Wasserstein ball centered around the original input distribution.

In FL, only a few works have explored the Wasserstein distance (Reisizadeh et al., 2020; Diamandis et al., 2021). Specially, Reisizadeh et al. (2020) proposed FedRobust based on adversarial robust training to enhance robustness. Regarding distributionally robust learning, Deng et al. (2020b) proposed DRFA, a communication efficient distributed algorithm. Besides, based on the agnostic FL framework suggested by Mohri et al. (2019), Du et al. (2020) introduced AgnosticFair, a two-player adversarial minimax game between the learner and the adversary, to achieve fairness.

## 3 WASSERSTEIN ROBUST FEDERATED LEARNING

### 3.1 EXPECTED RISK AND EMPIRICAL RISK MINIMIZATION IN FEDERATED LEARNING

In a federated setting, there are $m$ clients, and each client $i \in [m] := \{1, \ldots, m\}$ has its data generating distribution $P_i$ supported on domain $\mathcal{Z}_i := (\mathcal{X}_i, \mathcal{Y}_i)$. Consider the parametrized hypothesis class $\mathcal{H} = \{h_\theta \mid \theta \in \mathbb{R}^d\}$, where each member $h_\theta$ is a mapping from $\mathcal{X}_i$ to $\mathcal{Y}_i$ parametrized by $\theta$. With $z_i := (x_i, y_i) \in \mathcal{Z}_i$, we use $\ell(z_i, h_\theta)$, shorthand for $\ell(y_i, h_\theta(x_i))$, to represent the cost of predicting $h_\theta(x_i)$ when the ground-truth label is $y_i$. For example, if $h_\theta(x_i) = \theta^\mathsf{T} x_i$ and $y_i \in \mathbb{R}$, a square loss $\ell(z_i, h_\theta) = \ell(y_i, h_\theta(x_i)) = (\theta^\mathsf{T} x_i - y_i)^2$ can be considered. In FL, all clients collaborate with a server to find a global model $\theta$ such that the following sum weighted risk is minimized:

$$\min_{\theta \in \mathbb{R}^d} \sum_{i=1}^m \lambda_i \mathbf{E}_{Z_i \sim P_i} \big[ \ell(Z_i, h_\theta) \big], \tag{1}$$

where $\mathbf{E}_{Z_i \sim P_i} \big[ \ell(Z_i, h_\theta) \big]$ is client $i$'s expected risk and $\lambda_i \geq 0$ represents the relative "weight" of client $i$ and $\sum_{i=1}^m \lambda_i = 1$. Therefore, $\lambda := [\lambda_1, \ldots, \lambda_m]^\top$ belongs to a simplex $\Delta := \big\{ \lambda \in \mathbb{R}^m : \lambda \succcurlyeq 0 \text{ and } \lambda^\top \mathbf{1}_m = 1 \big\}$. Define by $P_\lambda := \sum_{i=1}^m \lambda_i P_i$ the mixed clients' distribution over $m$ domains $\mathcal{Z} := \{\mathcal{Z}_1, \ldots, \mathcal{Z}_m\}$. We denote by $Z \sim P_\lambda$ a random data point $Z$ generated by $P_\lambda$, which means that the domain of client $i$ is chosen with probability $\mathbf{P}(Z = Z_i) = \lambda_i$ first, then a data point $z_i \in \mathcal{Z}_i$ is selected with probability $\mathbf{P}(Z_i = z_i), Z_i \sim P_i$.

While the underlying distributions $P_i$ are unknown, clients have access to finite observations $z_i \in [n_i] \sim P_\lambda$. We abuse the notation $[n_i]$ to denote the set of client $i$'s both observable data points and their indexes. Let $\widehat{P}_{n_i} := \frac{1}{n_i} \sum_{z_i \in [n_i]} \delta_{z_i}$ be the empirical distribution of $P_i$, where $\delta_{z_i}$ is the Dirac point mass at $z_i$. In general, we use the notation $\widehat{\phantom{x}}$ for quantities that are dependent on the training data. Define by $\widehat{P}_\lambda := \sum_{i=1}^m \lambda_i \widehat{P}_{n_i}$ the mixed empirical distribution of $n = \sum_{i=1}^m n_i$ training data from $m$ clients. The empirical risk minimization (ERM) problem of (1) is as follows:

$$\min_{\theta \in \mathbb{R}^d} \left\{ \mathbf{E}_{Z \sim \widehat{P}_\lambda} \big[ \ell(Z, h_\theta) \big] = \sum_{i=1}^m \lambda_i \mathbf{E}_{Z_i \sim \widehat{P}_{n_i}} \big[ \ell(Z_i, h_\theta) \big] = \sum_{i=1}^m \frac{n_i}{n} \left( \frac{1}{n_i} \sum_{z_i \in [n_i]} \ell(z_i, h_\theta) \right) \right\}, \tag{2}$$

where $\lambda_i = n_i/n$ is often chosen in ERM of the standard FL (McMahan et al., 2017).

### 3.2 WASSERSTEIN ROBUST RISK IN FEDERATED LEARNING

Models resulting from (2) have been shown to be vulnerable to adversarial attacks and to lack of robustness to distribution shifts. We consider a robust variant of the ERM framework, involving the worst-case risk with respect to the $p$-Wasserstein distance between two probability measures. Given a set $\mathcal{Z}$, define $d : \mathcal{Z} \times \mathcal{Z} \rightarrow [0, \infty)$ to be the cost of "transportation" between its two points.[1] Suppose $P$ and $Q$ are two distributions on $\mathcal{Z}$. Let $\Pi(P, Q)$ be the set of probability measures $\pi$ on $\mathcal{Z} \times \mathcal{Z}$ whose marginals are $P$ and $Q$, called their couplings. In other words, $\pi(A, \mathcal{Z}) = P(A)$ and $\pi(\mathcal{Z}, A) = Q(A), \forall A \subset \mathcal{Z}$. The *$p$-Wasserstein distance* between $P$ and $Q$ is defined as

$$W_p(P, Q) = \inf_{\pi \in \Pi(P,Q)} \big( \mathbf{E}_{(Z,Z') \sim \pi} \big[ d^p(Z, Z') \big] \big)^{1/p}. \tag{3}$$

This distance represents the minimum cost of transporting one distribution to another, where the cost of moving a unit point mass is determined by the ground metric on the space of uncertainty realizations. In this work, we mainly work with $p = 2$. Let $\mathcal{B}_p(P, \rho) := \{ Q : W_p(P, Q) \leq \rho \}$

---

[1]The function $d$ must satisfy non-negativity, lower semi-continuity and $d(z, z) = 0, \forall z \in \mathcal{Z}$.

denote the *Wasserstein ball* centered at $P$ (i.e., nomimal distribution) and having radius $\rho \geq 0$. We modify (2) into the following Wasserstein robust risk minimization in FL problem:

$$\text{WAFL:} \quad \min_{\theta \in \mathbb{R}^d} \left\{ \sup_{Q \in \mathcal{B}(\widehat{P}_\lambda, \rho)} \mathbf{E}_{Z' \sim Q} \left[ \ell(Z', h_\theta) \right] \right\}. \tag{4}$$

There are several merits to this framework. First, the *ambiguity set* $\mathcal{B}_p(\widehat{P}_\lambda, \rho)$ contains all (continuous or discrete) distributions $Q$ that can be converted from the (discrete) nominal distribution $\widehat{P}_\lambda$ at a bounded transportation cost $\rho$. Second, Wasserstein distances can be approximated from the samples. Based on the non-asymptotic convergence results of Fournier & Guillin (2015), we can specify a suitable value for $\rho$ to probabilistically bound $W_p(P, Q)$ by the distance between their empirical distributions $W_p(\widehat{P}, \widehat{Q})$ (e.g., for multi-source domain adaptation).

In any robust optimization problem, the ambiguity set is a key ingredient to defining the level of robustness. We will compare WAFL in (4) with other approaches in terms of their ambiguity set.

**Agnostic FL.** Using this approach, existing techniques (Mohri et al., 2019; Deng et al., 2020b) minimize the worst-case loss

$$\max_{\lambda \in \Delta} \mathbf{E}_{Z \sim \widehat{P}_\lambda} \left[ \ell(Z, h_\theta) \right],$$

hence its distributional ambiguity set is $\mathcal{Q}_\Delta := \left\{ \widehat{P}_\lambda : \lambda \in \Delta \right\}$. While Agnostic FL's ambiguity set is the static convex hull of $\left\{ \widehat{P}_{n_i} \right\}_{i \in [m]}$, WAFL's ambiguity set $\mathcal{B}(\widehat{P}_\lambda, \rho)$ can be adjusted by controlling the robustness level $\rho$ and by positioning the ball center using $\lambda$, which is useful for domain adaptation. Therefore, $\mathcal{B}(\widehat{P}_\lambda, \rho)$ can cover $\mathcal{Q}_\Delta$ by using appropriate values for $\rho$ and $\lambda$ (see Fig. 1).

**Adversarial robust FL.** Using this approach, Reisizadeh et al. (2020) combines a general affine covariate shift in standard adversarial robust training with FL. Most existing techniques using this approach (Goodfellow et al., 2015; Kurakin et al., 2017; Carlini & Wagner, 2017; Madry et al., 2019; Tramèr et al., 2020) define an adversarial perturbation $u$ at

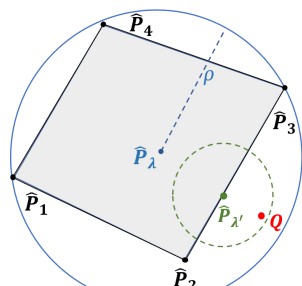

Figure 1: Example of four FL clients with data distributions $P_1, \ldots, P_4$. The shaded area (Agnostic FL's ambiguity set) is covered by the blue ball with radius $\rho$ and centered at $\widehat{P}_\lambda$ (Wasserstein ambiguity set). For domain adaptation with $Q$ as a target domain, the nominal distribution (multi-source domain) is shifted to $\widehat{P}_{\lambda'}$ such that $W_2(\widehat{P}_{\lambda'}, Q)$ is minimal.

a data point $Z$, and minimize the worst-case loss over all perturbations: $\max_{u \in \mathcal{U}} \mathbf{E}_{Z \sim \widehat{P}_\lambda} \left[ \ell(Z + u, h_\theta) \right]$, where the ambiguity set is $\mathcal{U} := \left\{ u \in \mathbb{R}^{d+1} : \|u\| \leq \epsilon \right\}$. In Appendix A, we show that the Wasserstein ambiguity set also contains the perturbation points induced by the solution to this adversarial robust training problem.

### 3.3 WAFL: Algorithm Design and Convergence Analysis

The original form of WAFL in (4) is not friendly for distributed algorithm design. Fortunately, the *Wasserstein robust risk* (or $\mathcal{B}(\widehat{P}_\lambda, \rho)$-worst-case risk) has its dual formulation as follows (Gao & Kleywegt, 2016; Sinha et al., 2020)

$$\sup_{Q \in \mathcal{B}(\widehat{P}_\lambda, \rho)} \mathbf{E}_{Z' \sim Q} \left[ \ell(Z', h_\theta) \right] = \inf_{\gamma \geq 0} \left\{ \gamma \rho^2 + \mathbf{E}_{Z \sim \widehat{P}_\lambda} \left[ \phi_\gamma(Z, \theta) \right] \right\}, \tag{5}$$

where $\phi_\gamma(z_i, \theta) := \sup_{\zeta \in \mathcal{Z}} \left[ \ell(\zeta, h_\theta) - \gamma d^2(\zeta, z_i) \right]$, and $d^2(z, z') = \|x - x'\|^2 + \kappa |y - y'|^2, \kappa > 0$. The crux of using the dual is that the inner supremum problem (finding $\phi_\gamma$) is easily solvable when its objective is well-conditioned: if $\ell$ is $L$-smooth and $d$ is 1-strongly convex, setting $\gamma > L$ ensures that $\zeta \mapsto \ell(\zeta, h_\theta) - \gamma d^2(\zeta, z)$ is strongly concave, and using gradient ascent for the inner supremum problem (for finding $\phi_\gamma$) ensures linear convergence. For each $z_i$ let $z_i^*$ be the solution to $\sup_{\zeta \in \mathcal{Z}} \left[ \ell(\zeta, h_\theta) - \gamma d^2(\zeta, z_i) \right]$. Then, $\nabla_\theta \phi_\gamma(z_i, \theta) = \nabla_\theta \ell(z_i^*, h_\theta)$ (Sinha et al., 2020, Lemma 1).

Rather than find the optimal $\gamma$ in (5), we consider $\gamma$ as a control parameter, and instead solve the following client-decomposable problem, which is amenable to distributed algorithm design.

$$\min_{\theta \in \mathbb{R}^d} \left\{ \mathbf{E}_{Z \sim \widehat{P}_\lambda} \left[ \phi_\gamma(Z, \theta) \right] = \sum_{i=1}^m \lambda_i \mathbf{E}_{Z_i \sim \widehat{P}_{n_i}} \left[ \phi_\gamma(Z_i, \theta) \right] \right\}. \tag{6}$$

---

**Algorithm 1:** Local SGD for WAFL

---

1: **for** $t = 0, 1, ..., T-1$ **do**                       `// Global rounds`
2:     Sample a subset of clients $S_t \subset [m]$
3:     **for** *client $i \in S_t$ in parallel* **do**
4:        Set local parameters: $\theta_i^{(t,0)} = \theta^t$             `// Communication`
5:        **for** $k = 0, 1, ..., K-1$ **do**              `// Local rounds`
6:           Sample a mini-batch $\mathcal{D}_i$ from client $i$'s dataset
7:           $\theta_i^{(t,k+1)} = \theta_i^{(t,k)} - \frac{\eta}{|\mathcal{D}_i|} \sum_{z_i \in \mathcal{D}_i} \nabla_\theta \phi(z_i, \theta_i^{(t,k)})$
8:        Send $\theta_i^{(t,K)}$ to server                   `// Communication`
9:     **Server:** update $\theta^{(t+1)} = \sum_{i \in S_t} \lambda_i \theta_i^{(t,K)} / \sum_{i \in S_t} \lambda_i$

---

This motivates the development of Algorithm 1 for solving (6). The structure of WAFL is similar to FedAvg with $T$ communication rounds. We further note three key components of Algorithm 1 common to FL methods. First, *client sampling* (line 2) refers to the partial participation of clients in each global round. Second, each client performs $K$ *local steps* (line 5) before sending its local model to the server,. Finally, *stochastic approximation* of a client's gradient using a mini-batch (lines 6 and 7) is necessary when the data size is large. The main difference between WAFL and FedAvg is that WAFL aims to minimize the risk with respect to the surrogate loss $\phi_\gamma$, rather than $\ell$.

We show that the convergence of WAFL can be similarly characterized as that of FedAvg, the de facto FL algorithm based on local SGD updates (McMahan et al., 2017). In FedAvg optimization, we seek to establish the convergence when using the original loss function $\ell$. On the other hand, in WAFL the convergence is with respect to the surrogate loss $\phi_\gamma$, through which the local and global risks are defined by $F_i(\theta) := \mathbf{E}_{Z_i \sim P_i}[\phi_\gamma(Z_i, \theta)]$ and $F(\theta) := \mathbf{E}_{Z \sim P_\lambda}[\phi_\gamma(Z, \theta)]$, respectively.

We first make the following assumptions, common to analyses of Wasserstein-robust optimization (Sinha et al., 2020). Unless stated otherwise, all norms are the Euclidean norm.

**Assumption 1.** *The function $d : \mathcal{Z} \times \mathcal{Z} \to \mathbb{R}_+$ is continuous, and $d(\cdot, z_0)$ is 1-strongly convex, $\forall z_0 \in \mathcal{Z}$.*

**Assumption 2.** *The loss function $\ell : \mathcal{Z} \times \mathcal{H} \to \mathbb{R}$ is Lipschitz continuous as follows*

$$(a)\|\ell(z, h_\theta) - \ell(z', h_\theta)\| \le L_z \|z - z'\|, \quad (b)\|\ell(z, h_\theta) - \ell(z, h_{\theta'})\| \le L_\theta \|\theta - \theta'\|.$$

**Assumption 3.** *The loss function $\ell : \mathcal{Z} \times \mathcal{H} \mapsto \mathbb{R}$ is Lipschitz smooth as follows*

$$\|\nabla_\theta \ell(z, h_\theta) - \nabla_\theta \ell(z, h_{\theta'})\| \le L_{\theta\theta}\|\theta - \theta'\|, \quad \|\nabla_z \ell(z, h_\theta) - \nabla_z \ell(z', h_\theta)\| \le L_{zz}\|z - z'\|,$$
$$\|\nabla_\theta \ell(z, h_\theta) - \nabla_\theta \ell(z', h_\theta)\| \le L_{\theta z}\|z - z'\|, \quad \|\nabla_z \ell(z, h_\theta) - \nabla_z \ell(z, h_{\theta'})\| \le L_{z\theta}\|\theta - \theta'\|.$$

Given Assumption 3, it has been shown that the mapping $\theta \mapsto \phi_\gamma(\cdot, \theta)$ is $L$-smooth with $L = L_{\theta\theta} + \frac{L_{\theta z} L_{z\theta}}{\gamma - L_{zz}}, \gamma > L_{zz}$ (Sinha et al., 2020) (more detail in Appendix Lemma 2). In addition, we make the following assumptions common to FL analysis (Wang et al., 2021).

**Assumption 4.** *The unbiased stochastic approximation of $\nabla F_i(\theta)$, denoted by $g_{\phi_i}(\theta) := \nabla_\theta \phi_\gamma(z_i, \theta), z_i \sim P_i$, has $\sigma^2$-uniformly bounded variance, i.e., $\mathbf{E}\left[\|g_{\phi_i}(\theta) - \nabla F_i(\theta)\|^2\right] \le \sigma^2$.*

**Assumption 5.** *The difference between the local gradient $\nabla F_i(\theta)$ and the global gradient $\nabla F(\theta)$ is $\Omega$-uniformly bounded, i.e., $\max_i \sup_\theta \|\nabla F_i(\theta) - \nabla F(\theta)\| \le \Omega$.*

Assuming complete participation of clients in every round ($S_t = m, \forall t$), using standard techniques in Wang et al. (2021), we have:

**Theorem 1** (WAFL's convergence for convex loss function)**.** *Let Assumptions 1–5 hold and the mapping $\theta \mapsto \ell(z, h_\theta)$ be convex. Denote by $\bar{\theta}^{(t,k)}$ the "shadow" sequence, defined as $\bar{\theta}^{(t,k)} = \sum_{i=1}^m \lambda_i \theta_i^{(t,k)}$ and by $\theta^*$ the optimal solution to $\min_{\theta \in \mathbb{R}^d} F(\theta)$. If the client learning rate satisfies*

$$\eta \le \min\left\{\frac{1}{3L}, \frac{D}{2\sqrt{KT\Lambda\hat{\sigma}}}, \frac{D^{\frac{2}{3}}}{48^{\frac{1}{3}} K^{\frac{2}{3}} T^{\frac{1}{3}} L^{\frac{1}{3}} \bar{\Omega}^{\frac{2}{3}}}, \frac{D^{\frac{2}{3}}}{40^{\frac{1}{3}} KT^{\frac{1}{3}} L^{\frac{1}{3}} \bar{\Omega}^{\frac{2}{3}}}\right\},$$

*then we have*

$$\mathbb{E}\left[\frac{1}{KT}\sum_{t=0}^{T-1}\sum_{k=0}^{K-1}F(\bar{\theta}^{(t,k)})-F(\theta^*)\right]\leq\mathcal{O}\left(\frac{LD^2}{KT}+\frac{\widehat{\sigma}D\Lambda^{\frac{1}{2}}}{\sqrt{KT}}+\frac{L^{\frac{1}{3}}\bar{\Omega}^{\frac{2}{3}}D^{\frac{4}{3}}}{K^{\frac{1}{3}}T^{\frac{2}{3}}}+\frac{L^{\frac{1}{3}}\bar{\Omega}^{\frac{2}{3}}D^{\frac{4}{3}}}{T^{\frac{2}{3}}}\right),\quad(7)$$

*where $\hat{\sigma}^2:=\frac{\sigma^2}{|\mathcal{D}_i|}$, $D:=\|\theta^{(0)}-\theta^*\|$ and $\Lambda:=\sum_{i=1}^m\lambda_i^2$.*

We provide a proof of Theorem 1 in Appendix C.

## 4 ROBUST GENERALIZATION BOUNDS

We show the generalization and robustness properties of WAFL's output by bounding its excess risk. Denote the loss class by $\mathcal{F}:=\ell\circ\mathcal{H}:=\{z\mapsto\ell(z,h),h\in\mathcal{H}\}$, where we use $f$ (resp. $f_\theta$) $\in\mathcal{F}$ to represent a generic loss (resp. a loss function parametrized by $\theta$).

**Definition 1.** Denote the expected risk and surrogate of Wasserstein robust risk of an arbitrary $f$, respectively, as follows

$$\mathcal{L}(P_\lambda,f):=\mathbf{E}_{Z\sim P_\lambda}\big[\ell(Z,h)\big]\quad\text{and}\quad\mathcal{L}_\rho^\gamma(P_\lambda,f):=\mathbf{E}_{Z\sim P_\lambda}\big[\phi_\gamma(Z,f)\big]+\gamma\rho^2.$$

Then their excess risks are defined respectively as follows

$$\mathcal{E}(P_\lambda,f):=\mathcal{L}(P_\lambda,f)-\inf_{f'\in\mathcal{F}}\mathcal{L}(P_\lambda,f')\quad\text{and}\quad\mathcal{E}_\rho^\gamma(P_\lambda,f):=\mathcal{L}_\rho^\gamma(P_\lambda,f)-\inf_{f'\in\mathcal{F}}\mathcal{L}_\rho^\gamma(P_\lambda,f').$$

If a distribution $Q$ is in the ambiguity set $\mathcal{B}(P_\lambda,\rho)$, we can bound its excess risk $\mathcal{E}(Q,f)$ as follows.

**Lemma 1.** *Let Assumption 2 (a) holds and $\gamma\geq L_z/\rho$. For all $f\in\mathcal{F}$ and for all $Q\in\mathcal{B}(P_\lambda,\rho)$,*

$$\mathcal{E}_\rho^\gamma(P_\lambda,f)-g(\rho,\gamma)\leq\mathcal{E}(Q,f)\leq\mathcal{E}_\rho^\gamma(P_\lambda,f)+g(\rho,\gamma),$$

*where $g(\rho,\gamma):=2L_z\rho+|\gamma-\gamma^*|\rho^2$, and $\gamma^*:=\arg\min_{\gamma'\geq0}\mathcal{L}_\rho^{\gamma'}(P_\lambda,f)$.*

We provide a proof of Lemma 1 in Appendix D.

**Remark 1.** Lemma 1 shows that the lower and upper bounds for $\mathcal{E}(Q,f)$ can be analyzed using $\mathcal{E}_\rho^\gamma(P_\lambda,f)$ and a two-component error term $g(\rho,\gamma)$. The first component, $2L_z\rho$, says that when $\rho$ is increased – to allow for larger Wasserstein distance between the nominal $P_\lambda$ and any worst-case distribution $Q$ – the difference between the excess risks $\mathcal{E}(Q,f)$ and $\mathcal{E}_\rho^\gamma(P_\lambda,f)$ increases, and this error is amplified at most by the Lipschitz constant $L_z$ of the mapping $z\mapsto\ell(z,\cdot)$. The second component, $|\gamma-\gamma^*|\rho^2$, addresses the sub-optimality error of a chosen value of $\gamma$, which is amplified when $\gamma$ is drifted away from the optimal $\gamma^*$. Note that $\mathcal{L}_\rho^{\gamma^*}(P_\lambda,f)$ is the same as $\mathcal{B}(P_\lambda,\rho)$-worst-case risk thanks to the strong duality (5), which is obtained with $\rho>0$.

Denote by $\widehat{\theta}^\epsilon\in\Theta$ an $\varepsilon$-minimizer to the surrogate ERM, i.e., $\mathbf{E}_{Z\sim\widehat{P}_\lambda}\big[\phi_\gamma(Z,f_{\widehat{\theta}^\varepsilon})\big]\leq\inf_{\theta\in\Theta}\mathbf{E}_{Z\sim\widehat{P}_\lambda}\big[\phi(Z,f_\theta)\big]+\varepsilon$, where $\Theta\subset\mathbb{R}^d$ is a parameter class, we obtain the following result.

**Theorem 2** (Robust Generalization Bounds). *Let Assumptions 2 and 3 hold, $\gamma\geq\max\{L_{zz},L_z/\rho\}$, and $|\ell(z,h)|\leq M_\ell$. We have the following result for all $Q\in\mathcal{B}(P_\lambda,\rho)$*

$$\mathcal{E}(Q,f_{\widehat{\theta}^\varepsilon})\leq\sum_{i=1}^m\lambda_i\left[\frac{48\mathcal{C}(\Theta)}{\sqrt{n_i}}+2M_\ell\sqrt{\frac{2\log(2m/\delta)}{n_i}}\right]+\varepsilon+g(\rho,\gamma)$$

*with probability at least $1-\delta$, where $\mathcal{C}(\Theta):=L_\theta\int_0^\infty\sqrt{\log\mathcal{N}\left(\Theta,\|\cdot\|_\Theta,\epsilon\right)}d\epsilon$ and $\mathcal{N}\left(\Theta,\|\cdot\|_\Theta,\epsilon\right)$ denotes the $\epsilon$-covering number of $\Theta$ w.r.t a metric $\|\cdot\|_\Theta$ as the norm on $\Theta$.*

We provide a proof of Theorem 2 in Appendix E. We sketch the proof as follows: first bound $\mathcal{E}_\rho^\gamma(P_\lambda,f_{\widehat{\theta}^\varepsilon})$ using standard excess risk decomposition and uniform convergence with Rademacher complexity. Then leverage the upper-bound of Lemma 1 to bound $\mathcal{E}(Q,f),\forall Q\in\mathcal{B}(P_\lambda,\rho)$, based on the bound of $\mathcal{E}_\rho^\gamma(P_\lambda,f_{\widehat{\theta}^\varepsilon})$. The result shows that using WAFL to minimize the surrogate of Wasserstein robust empirical risk also controls the robustness and generalization. As an example

$\mathcal{H} = \left\{ \langle \theta, \cdot \rangle, \theta \in \Theta \right\}$ with $\Theta = \left\{ \theta \in \mathbb{R}^d : \|\theta\|_2 \leq C \right\}$. The diameter of $\Theta$ is $\sup_{\theta, \theta' \in \Theta} \|\theta - \theta'\| = 2C$, thus $\mathcal{N}(\Theta, \|\cdot\|_2, \epsilon) = (1 + 2C/\epsilon)^d$, and $\mathcal{C}(\Theta) \leq 3CL_\theta \sqrt{d}/2$ (Lee & Raginsky, 2018).

Generally, the radius of Wasserstein ball $\rho$ can be considered hyperparameter that needs fine-tuning (e.g., through cross-validation). In principle, $\rho$ should not be too large to become over-conservative, which can hurt the empirical average performance, but also not too small to become similar to the ERM, and thus can lack robustness. From a statistical standpoint, we are interested in learning how to scale $\rho$ w.r.t. the sample size $n_i, i \in [m]$, such that the generalization of the WAFL solution $\widehat{\theta}^{\varepsilon}$ w.r.t the true distribution $P_\lambda$ is guaranteed, while still ensuring robustness w.r.t all distributions inside the Wasserstein ball. Using the result from Fournier & Guillin (2015), which shows that $\widehat{P}_{n_i}$ converges in Wasserstein distance to the true $P_i$ at a specific rate, we obtain the following

**Corollary 1.** *With all assumptions as in Theorem 2, defining $\rho_n := \sqrt{\sum_{i=1}^m \lambda_i \widehat{\rho}_{n_i}^{\delta/m}}$, we have*

$$\mathcal{E}(P_\lambda, f_{\widehat{\theta}^{\varepsilon}}) \leq \sum_{i=1}^m \lambda_i \left[ \frac{48\mathcal{C}(\Theta)}{\sqrt{n_i}} + 2M_\ell \sqrt{\frac{2\log(4m/\delta)}{n_i}} \right] + g(\rho_n, \gamma) + \varepsilon$$

*with probability at least $1 - \delta$, where* $\widehat{\rho}_n^\delta := \begin{cases} \left( \frac{\log(c_1/\delta)}{c_2 n} \right)^{\min\{2/d, 1/2\}} & \text{if } n \geq \frac{\log(c_1/\delta)}{c_2}, \\ \left( \frac{\log(c_1/\delta)}{c_2 n} \right)^{1/\alpha} & \text{if } n < \frac{\log(c_1/\delta)}{c_2}. \end{cases}$

We provide a proof of Corollary 1 in Appendix F.

## 5 Choosing $\lambda$: Applications

We focus on two applications: multi-source domain adaptation and generalization to all client distributions. We provide insights about choosing the client weights $\lambda$ for these applications.

**Multi-source domain adaptation:** Consider the multi-source domain distribution $P_\lambda$ (Mansour et al., 2021). Lee & Raginsky (2018) show that solving the minimax risk with the Wasserstein ambiguity set can help transfer data/knowledge from the source domain $P_\lambda$ to a different, but related, target domain $Q$. They bound the distance $W_p(P_\lambda, Q)$ using the triangle inequality

$$W_p(P_\lambda, Q) \leq W_p(P_\lambda, \widehat{P}_\lambda) + W_p(\widehat{P}_\lambda, \widehat{Q}) + W_p(\widehat{Q}, Q), \tag{8}$$

where $\widehat{P}_\lambda$ and $\widehat{Q}$ are the empirical versions of $P_\lambda$ and $Q$, respectively. While $W_p(P_\lambda, \widehat{P}_\lambda)$ and $W_p(\widehat{Q}, Q)$ can be probabilistically bounded with a confidence parameter $\delta \in (0, 1)$ according to Fournier & Guillin (2015), $W_p(\widehat{P}_\lambda, \widehat{Q})$ can be deterministically computed using linear or convex programming (Peyré & Cuturi, 2019).
In FL context, in order to have a better bound for $W_2(P_\lambda, Q)$ similar to (8), it is straightforward to choose $\lambda = \arg\min_{\lambda' \in \Delta} W_2(\widehat{P}_{\lambda'}, \widehat{Q})$. To relax this problem into a form solvable using existing approaches, observe that $W_2(\widehat{P}_\lambda, Q) \leq \sum_{i=1}^m \lambda_i W_2(\widehat{P}_{n_i}, Q)$ due to the convexity of Wasserstein distance. We then consider the following upper-bound to $\min_{\lambda \in \Delta} W_2(\widehat{P}_\lambda, \widehat{Q})$:

$$\min_{\lambda \in \Delta} \sum_{i=1}^m \lambda_i W_2(\widehat{P}_{n_i}, \widehat{Q}) =: \rho^\star, \tag{9}$$

which is a linear program, considering each $W_2(\widehat{P}_{n_i}, \widehat{Q})$ can be found by efficiently solving convex programs especially with entropic regularization and the Sinkhorn algorithm (Cuturi, 2013).

**Corollary 2.** *Denote the solution to (9) by $\lambda^\star$, and assume that domain $Q$ generates $n_Q$ i.i.d. data points. With probability at least $1 - \delta$, we have*

$$W_2(P_{\lambda^\star}, Q) \leq W_2(P_{\lambda^\star}, \widehat{P}_{\lambda^\star}) + W_2(\widehat{P}_{\lambda^\star}, \widehat{Q}) + W_2(Q, \widehat{Q}) \leq \sqrt{\sum_{i=1}^m \lambda_i^\star \widehat{\rho}_{n_i}^{\delta/m}} + \rho^\star + \widehat{\rho}_{n_Q}^{\delta/2}.$$

The proof of this corollary is similar to that of Corollary 3 in Appendix B.

**Covering all client distributions in the Wasserstein ball:** Suppose we want to cover all client distributions inside a Wassertein ball so that the generalization and robustness result by WAFL in Theorem 2 is applicable to all clients' distributions. We show in Appendix B that this is a problem of finding $\lambda$ such that the Wasserstein distance between $P_\lambda$ and $P_j, \forall j$, is as small as possible.

## 6    EXPERIMENTS

We first show how to change the level of worst-case perturbations by varying the robust parameter $\gamma$. To show the generalizability and robustness of WAFL, we evaluate WAFL under non-i.i.d. and distribution shift settings. We compare WAFL with baseline robust methods and non-robust FedAvg. Finally, in Appendix G.4, we show WAFL's capability in domain adaptation.

**Experimental settings.** We use two datasets in two non-i.i.d. settings. We distribute the first dataset – MNIST (Lecun et al., 1998) – to 100 clients and use a multinomial logistic regression model to model a convex setting. For the second dataset – CIFAR-10 (Krizhevsky, 2009) – we use 20 clients and a CNN model employed in McMahan et al. (2017) to model a non-convex setting. We set $\lambda_i = n_i/n$ as the client weights. In optimization, we randomly sample $S_t = 10$ clients to participate in training at each communication round. More detail can be found in Appendix G.1.

**Effect of $\gamma$ on the worst-case risk perturbations.** We first examine the relationship between the robust parameter $\gamma$ and the average worst-case perturbations $\hat{\rho}$, defined as $\hat{\rho}^2 = \mathbf{E}_{Z \sim \widehat{P}_\lambda} \left[ d^2(\widehat{Z}, Z) \right]$, where $\widehat{Z}$ is the adversarial perturbation of $Z$. As shown in Fig. 2, smaller values of $\gamma$ correspond to larger worst-case perturbations $\hat{\rho}$ on both the MNIST and CIFAR-10 datasets. For the rest of the experiment, rather than control $\hat{\rho}$ directly, we set $\gamma$ on the opposite direction to control the level of robustness.

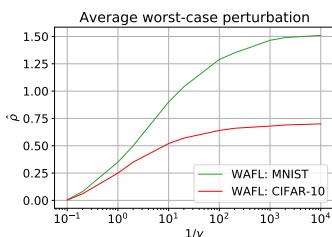

Figure 2: Varying $\hat{\rho}$ by $\gamma$.

**Generalization and robustness of WAFL.** We consider $\widehat{P}_\lambda$ and $\widehat{Q}$ as the empirical distribution of the training samples and test samples of all clients, respectively. By varying $\gamma$, we aim to train a global model robust to any empirical test distribution $\widehat{Q}$. To show the generalization and robustness of WAFL, we train and evaluate it in two different scenarios. First, in the *clean data* scenario, the global robust model is trained with different values of $\gamma$ and then evaluated on given clients' test data (similar to traditional FL). Second, in the *distribution shift* setting, the training process is similar; however, the global robust model is evaluated when there are distribution shifts at the clients' test data. To obtain these shifts, we use the common PGD attack method in Madry et al. (2019) under $l_\infty$-norm to generate an $\epsilon$-level perturbation on clients' test data. We choose the $l_\infty$-norm as it shows benefits in adversaries and gives large perturbations. Following Madry et al. (2019), we fix the number of gradient steps to generate adversarial examples with $t_{avd} = 40$, $\epsilon = 0.3$, $\alpha = 0.01$ for MNIST and $t_{avd} = 10$, $\epsilon = 8/255$, $\alpha = 2/255$ for CIFAR-10 with a batch size of 64. We note that this setting is similar to the adversarial poisoning attacks and the main purpose of this setting is to increase the Wasserstein distance between $\widehat{P}_\lambda$ and $\widehat{Q}$, thereby verifying the robustness of WAFL.

The performance of WAFL in both scenarios on MNIST and CIFAR-10 is shown in Fig. 3. When the clients' data is clean, the Wasserstein distance between $\widehat{P}_\lambda$ and $\widehat{Q}$ is relatively small, and training WAFL with small $\gamma$ (large $\hat{\rho}$) gives the worse performance on the test set. With a sufficiently large $\gamma$, WAFL is less robust and has the same generalization with FedAvg. By carefully fine-turning $\gamma$ in the range $[0.5, 1]$ for MNIST and $[10, 20]$ for CIFAR-10, WAFL shows an improvement over FedAvg. $\gamma$ in this scenario plays the same role as a regularization parameter to handle non-i.i.d. data.

By adding distribution shifts, we increase the Wasserstein distance between $\widehat{P}_\lambda$ and $\widehat{Q}$. By varying $\gamma$, we train a global robust model with different levels of worst-case perturbation to handle the distribution shifts. Small values of $\gamma$ generate larger ambiguity sets $\mathcal{B}(\widehat{P}_\lambda, \hat{\rho})$ and increase the robustness of WAFL, thus increase the chance $\widehat{Q}$ lies inside $\mathcal{B}(\widehat{P}_\lambda, \hat{\rho})$. In Fig. 3, WAFL shows better performance with smaller $\gamma$ values. However, as in mentioned in Sec. 4, when $\hat{\rho}$ is much larger or smaller than the level of distribution shifts ($\gamma \leq 0.01$ or $\gamma \geq 1$ for MNIST and $\gamma \leq 0.1$ or $\gamma > 10$ for CIFAR-10), WAFL performs inefficiently: too small $\gamma$ hurts the empirical performance, while too large $\gamma$ makes WAFL lack robustness. For every scenario, $\gamma$ needs to be tuned correspondingly. In our experiments, by carefully choosing $\gamma$, WAFL not only handles distribution shifts or common data poisoning attacks but also provides better performance than FedAvg in non-i.i.d. data and heterogeneous settings. Specifically, we set $\gamma = 0.5$ for MNIST and $\gamma = 10$ for CIFAR-10.

**Comparison with other robust methods.** We compare WAFL with three baselines: two common robust methods in FL called FedPGM and FedFGSM, and one non-robust method Fe-

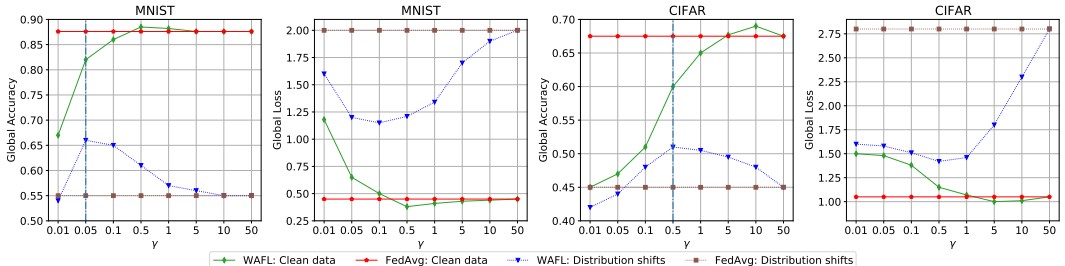

Figure 3: Global accuracy and loss of with different values of $\gamma$ on MNIST and CIFAR-10 under clean data and distribution shifts (40% of clients are affected by PGD attack). The blue vertical line indicates the value of $\gamma$ giving the same level $\epsilon$ of PGD attack($\gamma = 0.05$ for MNIST and $\gamma = 0.5$ for CIFAR-10).

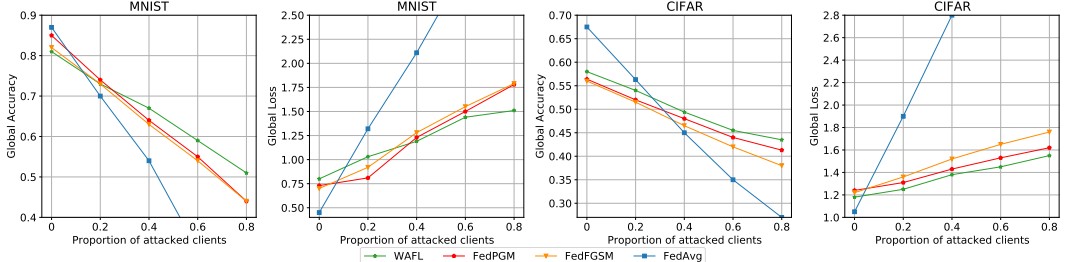

Figure 4: Comparison with other robust methods on MNIST and CIFAR-10 at different proportion of attacked clients (clients are affected by distribution shifts).

dAvg. FedPGM and FedFGSM are FedAvg with adversarial training using the projected gradient method PGD (Madry et al., 2019) and the fast-gradient method FGSM (Goodfellow et al., 2015), respectively. In FedPGM and FedFGSM, in each local update, all clients solve $\delta^* = \arg\max_{\|\delta\|_\infty \le \epsilon} \left\{ \ell(h_\theta(z + \delta), y) \right\}$ using projection onto an $l_\infty$-norm to find the worst-case perturbation $\delta$. While FedPGD uses $t_{avd}$ gradient steps to find $\delta^*$, FedFGSM uses only one gradient step. We use the same values of $\epsilon$ and $\alpha$ from the distribution shifts setting for projection. For a fair comparison, both WAFL and FedPGM have the same value $t_{avd}$ and all algorithms have the same number of local updates $K$. We also train WAFL with the value of $\gamma$ generating the same level perturbation of $\epsilon$ in FedPGM and FedFGSM. We study different proportions of clients having distribution shifts in Fig. 4. The robust accuracy of all methods decreases when the percentage of clients having distribution shifts increases. Especially, the FedAvg's performance drops dramatically when the percentage of attacked clients reaches 80%. For all scenarios, WAFL outperforms all the baselines. Specifically, in the case of 80% attacked clients, the improvements in accuracy of WAFL over FedPGM, FedFGSM, and FedAvg are 7%, 8%, and 33% for MNIST and 2.5%, 4.5%, and 18% for CIFAR-10, respectively.

## 7 CONCLUSION

In this paper, we present WAFL, a Wasserstein distributionally robust optimization framework, to tackle the issue of statistical heterogeneity in federated learning. We first remodel the duality of the worst-case risk to an empirical surrogate risk minimization problem, then solve it using a local SGD-based algorithm with convergence analysis. We show that WAFL is more general in terms of robustness compared to related approaches, and obtains an explicit robust generalization bound with respect to all unknown distributions in the Wasserstein ambiguity set. Through numerical experiments, we demonstrate that WAFL generalizes better than the standard FedAvg baseline in non-i.i.d. settings, and outperforms related methods with respect to robustness to distribution shifts. Moreover, WAFL reveals its capability in generalizing to unseen data distributions.

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

## A   ADVERSARIAL ROBUST FL'S AMBIGUITY SET V.S. WASSERTEIN BALL

We show that using the Wasserstein ambiguity set contains the perturbation points induced by the solution to the *Adversarial Robust FL* approach. As we present in Sec. 3.2, existing techniques for adversarial training robust models (Goodfellow et al., 2015; Papernot et al., 2015; Kurakin et al., 2017; Carlini & Wagner, 2017; Madry et al., 2019; Tramèr et al., 2020) define an adversarial perturbation $u$ at a data point $Z$, and minimize the following worst-case loss over all possible perturbations

$$\max_{u \in \mathcal{U}} \mathbf{E}_{Z \sim \widehat{P}_\lambda} \Big[ \ell(Z + u, h_\theta) \Big], \tag{10}$$

where the ambiguity set $\mathcal{U} := \big\{ u \in \mathbb{R}^{d+1} : \|u\| \leq \epsilon \big\}$. To compare this approach with Wasserstein-robust FL, we relate the above problem to its counterpart defined in the probability space of input as follows

$$\max_{\tilde{Q} \in \mathcal{Q}(\epsilon)} \mathbf{E}_{\tilde{Z} \sim \tilde{Q}} \Big[ \ell(\tilde{Z}, h_\theta) \Big], \tag{11}$$

where $\mathcal{Q}(\epsilon) := \Big\{ \tilde{Q} : \mathbb{P} \big[ \|\tilde{Z} - Z\| \leq \epsilon \big] = 1, Z \sim \widehat{P}_\lambda, \tilde{Z} \sim \tilde{Q} \Big\}$. Considering $u^*$ as a solution to problem (10), we see that the distribution $Q'$ of perturbation points (i.e., $\tilde{Z} := (Z + u^*) \sim Q'$) in problem (10) belongs to the feasible set $\mathcal{Q}(\epsilon)$ in problem (11) (If not, then $\mathbb{P} \big[ \|u^*\| \leq \epsilon \big] < 1$, a contradiction). Next, consider an arbitrary distribution $\tilde{Q} \in \mathcal{Q}(\epsilon)$ in problem (11), with any $\tilde{Z} \sim \tilde{Q}$ and $Z \sim \widehat{P}_\lambda$, we have

$$\|\tilde{Z} - Z\| \overset{\text{w.p.1}}{\leq} \epsilon \implies \mathbf{E}_{Z \sim \widehat{P}_\lambda, \tilde{Z} \sim \tilde{Q}} \big[ \|\tilde{Z} - Z\| \big] \leq \epsilon \implies \inf_{\pi \in \Pi(\widehat{P}_\lambda, \tilde{Q})} \mathbf{E}_{(Z, Z') \sim \pi} \big[ \|\tilde{Z} - Z\| \big] \leq \epsilon,$$

which implies that $W_1(\widehat{P}_\lambda, \tilde{Q}) \leq \epsilon, \forall \tilde{Q} \in \mathcal{Q}(\epsilon)$, and thus $\mathcal{Q}(\epsilon) \subset \mathcal{B}_1(\widehat{P}_\lambda, \epsilon)$. We have shown that the Wasserstein ambiguity set contains the perturbation points induced by the solution to the adversarial robust training problem (10).

## B   CHOOSING $\lambda$: GENERALIZING TO ALL CLIENT DISTRIBUTIONS

We show that by calibrating appropriate $\lambda$ value, our proposed algorithm will be capable of generalizing to all client distributions. Suppose we want to cover all client distributions inside a Wassertein ball so that the generalization and robustness result by WAFL in Theorem 2 is applicable to all clients' distributions. This is the problem of finding $\lambda$ such that the Wasserstein distance between $P_\lambda$ and $P_j, \forall j$, is as small as possible. Instead of directly finding the minimum Wasserstein radius that cover all client distributions, we will leverage the popular Wasserstein barycenter problem (Peyré & Cuturi, 2019). Specifically, consider the problem

$$\min_{\lambda \in \Delta} W_2(\widehat{P}_\lambda, P^\heartsuit) \quad \text{s.t.} \quad P^\heartsuit = \arg\min_{Q \in \mathcal{P}} \sum_{i=1}^m \lambda_i W_2(\widehat{P}_{n_i}, Q), \tag{12}$$

where $P^\heartsuit$ is the Wasserstein bary center w.r.t the solution $\lambda^*$ to this problem. Even though the solution is not straightforward, we propose to solve its tractable upper-bound:

$$\min_{\lambda \in \Delta, Q \in \mathcal{P}} \sum_{i=1}^m \lambda_i W_2(\widehat{P}_{n_i}, Q). \tag{13}$$

This is a bi-convex problem, which is convex w.r.t to $\lambda$ (resp. $Q$) when fixing $Q$ (resp. $\lambda$). Thus, we can use alternative minimization (Gorski et al., 2007) to find a local solution to this problem. Denoting $\tilde{\lambda}$ as the solution to (12) and $(\lambda^*, P^*)$ as a local solution to (13), we obtain

$$W_2(\widehat{P}_{\tilde{\lambda}}, P^\heartsuit) \leq W_2(\widehat{P}_{\lambda^*}, P^*) \leq \sum_{i=1}^m \lambda_i^* W_2(\widehat{P}_{n_i}, P^*) =: \rho^*. \tag{14}$$

**Corollary 3.** *For all client $j \in [m]$, with probability at least $1 - \delta$, we have*

$$W_2(P_{\lambda^*}, P_j) \leq W_2(P_{\lambda^*}, \widehat{P}_{\lambda^*}) + W_2(\widehat{P}_{\lambda^*}, P^*) + W_2(P^*, \widehat{P}_{n_j}) + W_2(\widehat{P}_{n_j}, P_j)$$

$$\leq \sqrt{\sum_{i=1}^m \lambda_i^* \widehat{\rho}_{n_i}^{\delta/m}} + \rho^* + \frac{\rho^*}{\lambda_j} + \widehat{\rho}_{n_j}^{\delta/2}$$

*Proof.* The first line is by triangle inequality. The second line is by following facts: (i) $\mathbf{P}\left[W_2(P_{\lambda^*}, \widehat{P}_{\lambda^*}) \geq \sqrt{\sum_{i=1}^m \lambda_i^* \widehat{\rho}_{n_i}^{\delta/m}}\right] \leq \delta/2$ according to (37), (ii) $W_2(P^*, \widehat{P}_{n_j}) = \frac{\lambda_j W_2(P^*, \widehat{P}_{n_j})}{\lambda_j} \leq \frac{\rho^*}{\lambda_j}$, and (iii) $\mathbf{P}\left[W_2(\widehat{P}_{n_j}, P_j) \geq \widehat{\rho}_{n_j}^{\delta/2}\right] \leq \delta/2$ according to (35), and (iv) using union bound. $\square$

## C   PROOF OF THEOREM 1

Our proof is based on the analysis of local SGD for FL presented in Wang et al. (2021).

Fix some $z_i$. Define $\varphi(\zeta, \theta; z_i) := \ell(\zeta, h_\theta) - \gamma d(\zeta, z_i)$. Since $\ell$ is $L_{zz}$-smooth and $d$ is 1-strongly convex, $\varphi(\zeta, \theta; z_i)$ is $(\gamma - L_{zz})$-strongly concave with respect to $\zeta$, given that $\gamma > L_{zz}$.

**Lemma 2.** *Let* $z_i^* = \arg\max_{\zeta \in \mathcal{Z}} \varphi(\zeta, \theta; z_i)$. *Therefore,* $\phi_\gamma(\theta; z_i) = \varphi(z_i^*, \theta; z_i)$. *Let* $\ell$ *satisfy Assumption 3. Then* $\phi_\gamma$ *is differentiable, and*

$$\|\nabla \phi_\gamma(z_i, \theta) - \nabla \phi_\gamma(z_i, \theta')\| \leq L\|\theta - \theta'\|,$$

*with* $L = L_{\theta\theta} + \frac{L_{\theta z} L_{z\theta}}{\gamma - L_{zz}}$ *when* $\gamma > L_{zz}$.

The proof can be found in (Sinha et al., 2020, Lemma 1). Lemma 2 implies that $\phi_\gamma$ is $L$-smooth.

Define $g_i(\theta) := \frac{1}{|\mathcal{D}_i|} \sum_{z_i \in \mathcal{D}_i} \nabla_\theta \phi_\gamma(z_i, \theta)$, then we have

$$\mathbf{E}\left[\|g_i(\theta) - \nabla F_i(\theta)\|^2\right] = \mathbf{E}\left[\left\|\frac{1}{|\mathcal{D}_i|} \sum_{z_i \in \mathcal{D}_i} \nabla_\theta \phi_\gamma(z_i, \theta) - \nabla F_i(\theta)\right\|^2\right]$$

$$\leq \frac{1}{|\mathcal{D}_i|} \mathbf{E}\left[\|g_{\phi_i}(\theta) - \nabla F_i(\theta)\|^2\right] \leq \frac{\sigma^2}{|\mathcal{D}_i|} := \widehat{\sigma}^2 \quad \text{(by Assumption 4.)} \tag{15}$$

With the shadow sequence $\bar{\theta}^{(t,k)} = \sum_{i=1}^m \lambda_i \theta_i^{(t,k)}$, we have

$$\bar{\theta}^{(t,k+1)} = \sum_{i=1}^m \lambda_i \theta_i^{(t,k+1)} = \sum_{i=1}^m \lambda_i \big(\theta_i^{(t,k)} - \eta g_i(\theta_i^{(t,k)})\big) = \bar{\theta}^{(t,k)} - \eta \sum_{i=1}^m \lambda_i g_i(\theta_i^{(t,k)}).$$

**Lemma 3.** *If the client learning rate satisfies* $\eta \leq \frac{1}{3L}$, *then*

$$\frac{1}{K} \sum_{k=0}^{K-1} \mathbf{E}\left[F(\bar{\theta}^{(t,k)}) - F(\theta^*)\right] \leq 2\eta\hat{\sigma}^2 \left(\sum_{i=1}^m \lambda_i^2\right) + L \sum_{i=1}^m \lambda_i \sum_{k=0}^{K-1} \frac{1}{K} \mathbf{E}\left[\|\theta_i^{(t,k)} - \bar{\theta}^{(t,k)}\|^2\right]$$

$$+ \frac{1}{2\eta K}\left(\|\theta^{(t)} - \theta^*\|^2 - \mathbf{E}\left[\|\theta^{(t+1)} - \theta^*\|^2\right]\right).$$

*Proof.* Since $\bar{\theta}^{(t,k+1)} = \bar{\theta}^{(t,k)} - \eta \sum_{i=1}^m \lambda_i g_i(\theta_i^{(t,k)})$, by parallelogram law

$$\sum_{i=1}^m \lambda_i \left\langle g_i(\theta_i^{(t,k)}), \bar{\theta}^{(t,k+1)} - \theta^* \right\rangle = \frac{1}{2\eta}\left(\|\bar{\theta}^{(t,k)} - \theta^*\|^2 - \|\bar{\theta}^{(t,k+1)} - \bar{\theta}^{(t,k)}\|^2 - \|\bar{\theta}^{(t,k+1)} - \theta^*\|^2\right).$$

$$\tag{16}$$

Fact: $F_i(\theta) = \mathbf{E}_{Z_i \sim P_i}\left[\phi_\gamma(Z_i, \theta)\right]$ is Lipschitz smooth with $L = L_{\theta\theta} + \frac{L_{\theta z} L_{z\theta}}{\gamma - L_{zz}}$ when $\gamma > L_{zz}$. With the assumption that $\theta \mapsto \ell(z, h_\theta)$ is convex, we have $\theta \mapsto F_i(\theta)$ is convex.

Since $F_i$ is convex and $L$-smooth,

$$F_i(\bar{\theta}^{(t,k+1)}) \leq F_i(\theta_i^{(t,k)}) + \left\langle \nabla F_i(\theta_i^{(t,k)}), \bar{\theta}^{(t,k+1)} - \theta_i^{(t,k)} \right\rangle + \frac{L}{2}\|\bar{\theta}^{(t,k+1)} - \theta_i^{(t,k)}\|^2$$

$$\leq F_i(\theta^*) + \left\langle \nabla F_i(\theta_i^{(t,k)}), \bar{\theta}^{(t,k+1)} - \theta^* \right\rangle + \frac{L}{2}\|\bar{\theta}^{(t,k+1)} - \theta_i^{(t,k)}\|^2$$

$$\leq F_i(\theta^*) + \left\langle \nabla F_i(\theta_i^{(t,k)}), \bar{\theta}^{(t,k+1)} - \theta^* \right\rangle + L\|\bar{\theta}^{(t,k+1)} - \bar{\theta}^{(t,k)}\|^2 + L\|\theta_i^{(t,k)} - \bar{\theta}^{(t,k)}\|^2. \tag{17}$$

From (16) and (17), we have

$$F(\bar{\theta}^{(t,k+1)}) - F(\theta^*) = \sum_{i=1}^{m} \lambda_i \Big( F_i(\bar{\theta}^{(t,k+1)}) - F(\theta^*) \Big)$$

$$\leq \sum_{i=1}^{m} \lambda_i \left\langle \nabla F_i(\theta_i^{(t,k)}) - g_i(\theta_i^{(t,k)}), \bar{\theta}^{(t,k+1)} - \theta^* \right\rangle + L\|\bar{\theta}^{(t,k+1)} - \bar{\theta}^{(t,k)}\|^2 + L\sum_{i=1}^{m} \lambda_i \|\theta_i^{(t,k)} - \bar{\theta}^{(t,k)}\|^2$$

$$+ \frac{1}{2\eta}\Big( \|\bar{\theta}^{(t,k)} - \theta^*\|^2 - \|\bar{\theta}^{(t,k+1)} - \bar{\theta}^{(t,k)}\|^2 - \|\bar{\theta}^{(t,k+1)} - \theta^*\|^2 \Big). \tag{18}$$

We have

$$\mathbf{E}\Big[\sum_{i=1}^{m} \lambda_i \left\langle \nabla F_i(\theta_i^{(t,k)}) - g_i(\theta_i^{(t,k)}), \bar{\theta}^{(t,k+1)} - \theta^* \right\rangle \Big]$$

$$= \mathbf{E}\Big[\sum_{i=1}^{m} \lambda_i \left\langle \nabla F_i(\theta_i^{(t,k)}) - g_i(\theta_i^{(t,k)}), \bar{\theta}^{(t,k+1)} - \bar{\theta}^{(t,k)} \right\rangle \Big] \quad (\text{since } \mathbf{E}\big[g_i(\theta_i^{(t,k)})\big] = \nabla F_i(\theta_i^{(t,k)}) \text{ given } \bar{\theta}^{(t,k)}, \theta^*)$$

$$\leq \frac{3}{2}\eta \cdot \mathbf{E}\Big[\|\sum_{i=1}^{m} \lambda_i \big(\nabla F_i(\theta_i^{(t,k)}) - g_i(\theta_i^{(t,k)})\big)\|^2\Big] + \frac{1}{6\eta}\mathbf{E}\Big[\|\bar{\theta}^{(t,k+1)} - \bar{\theta}^{(t,k)}\|^2\Big] \quad (\text{by Peter Paul inequality})$$

$$\leq 2\eta\hat{\sigma}^2 \Big(\sum_{i=1}^{m} \lambda_i^2\Big) + \frac{1}{6\eta}\mathbf{E}\Big[\|\bar{\theta}^{(t,k+1)} - \bar{\theta}^{(t,k)}\|^2\Big], \tag{19}$$

Plugging (19) back to the conditional expectation of (18), and noting that $\eta \leq \frac{1}{3L}$, we have

$$\mathbf{E}\Big[F(\bar{\theta}^{(t,k+1)}) - F(\theta^*)\Big] + \frac{1}{2\eta}\left( \mathbf{E}\Big[\|\bar{\theta}^{(t,k+1)} - \theta^*\|^2\Big] - \|\bar{\theta}^{(t,k)} - \theta^*\|^2 \right)$$

$$\leq 2\eta\hat{\sigma}^2\Big(\sum_{i=1}^{m} \lambda_i^2\Big) - \Big(\frac{1}{3\eta} - L\Big)\mathbf{E}\Big[\|\bar{\theta}^{(t,k+1)} - \bar{\theta}^{(t,k)}\|^2\Big] + L\sum_{i=1}^{m} \lambda_i \|\theta_i^{(t,k)} - \bar{\theta}^{(t,k)}\|^2$$

$$\leq 2\eta\hat{\sigma}^2\Big(\sum_{i=1}^{m} \lambda_i^2\Big) + L\sum_{i=1}^{m} \lambda_i \|\theta_i^{(t,k)} - \bar{\theta}^{(t,k)}\|^2$$

By convexity of $F$ and telescoping $k$ from $0$ to $K-1$, we have

$$\frac{1}{K}\sum_{k=0}^{K-1} \mathbf{E}\Big[F(\bar{\theta}^{(t,k)}) - F(\theta^*)\Big] \leq 2\eta\hat{\sigma}^2\Big(\sum_{i=1}^{m} \lambda_i^2\Big) + L\sum_{i=1}^{m} \lambda_i \sum_{k=0}^{K-1} \frac{1}{K}\mathbf{E}\Big[\|\theta_i^{(t,k)} - \bar{\theta}^{(t,k)}\|^2\Big]$$

$$+ \frac{1}{2\eta K}\Big(\|\bar{\theta}^{(t,0)} - \theta^*\|^2 - \mathbf{E}\Big[\|\bar{\theta}^{(t,K)} - \theta^*\|^2\Big]\Big).$$

Since $\bar{\theta}^{(t,0)} = \theta^{(t)}$ and $\bar{\theta}^{(t,K)} = \theta^{(t+1)}$, we complete the proof. $\qquad\square$

**Lemma 4** (Bounded client drift). *Assuming the client learning rate satisfies $\eta \leq \frac{1}{3L}$, we have*

$$\mathbf{E}\Big[\|\theta_i^{(t,k)} - \bar{\theta}^{(t,k)}\|^2\Big] \leq \eta^2(24K^2\bar{\Omega}^2 + 20K\bar{\Omega}^2).$$

*where $\bar{\Omega}^2 := \max\{\hat{\sigma}^2, \Omega^2\}$.*

*Proof.*

$$\mathbf{E}\left[\big\|\theta_1^{(t,k+1)} - \theta_2^{(t,k+1)}\big\|^2\right] = \mathbf{E}\left[\Big\|\theta_1^{(t,k)} - \theta_2^{(t,k)} - \eta\Big(g_1(\theta_1^{(t,k)}) - g_2(\theta_1^{(t,k)})\Big)\Big\|^2\right]$$

$$= \|\theta_1^{(t,k)} - \theta_2^{(t,k)}\|^2 - 2\eta\left\langle g_1(\theta_1^{(t,k)}) - \nabla F_1(\theta_1^{(t,k)}), \theta_1^{(t,k)} - \theta_2^{(t,k)}\right\rangle$$

$$- 2\eta\left\langle \nabla F_2(\theta_1^{(t,k)}) - g_2(\theta_2^{(t,k)}), \theta_1^{(t,k)} - \theta_2^{(t,k)}\right\rangle$$

$$- 2\eta\left\langle \nabla F_1(\theta_1^{(t,k)}) - \nabla F_2(\theta_2^{(t,k)}), \theta_1^{(t,k)} - \theta_2^{(t,k)}\right\rangle + \eta^2\|g_1(\theta_1^{(t,k)}) - g_2(\theta_2^{(t,k)})\|^2. \tag{20}$$

The second term (and similarly for the third term) is bounded as follows

$$
\begin{aligned}
&-\left\langle g_1(\theta_1^{(t,k)}) - \nabla F_1(\theta_1^{(t,k)}), \theta_1^{(t,k)} - \theta_2^{(t,k)} \right\rangle \\
&\leq \frac{1}{6\eta K}\|\theta_1^{(t,k)} - \theta_2^{(t,k)}\|^2 + \frac{3\eta K}{2}\|g_1(\theta_1^{(t,k)}) - \nabla F_1(\theta_1^{(t,k)})\|^2 \quad \text{(by Peter Paul inequality)} \\
&= \frac{1}{6\eta K}\|\theta_1^{(t,k)} - \theta_2^{(t,k)}\|^2 + \frac{3\eta K}{2}\widehat{\sigma}^2 \quad \text{(by (15))}
\end{aligned}
$$

Since $\max_i \sup_\theta \|\nabla F_i(\theta) - \nabla F(\theta)\| \leq \Omega$ (Assumption 5), the 4th-term is bounded as

$$
\begin{aligned}
&-\left\langle \nabla F_1(\theta_1^{(t,k)}) - \nabla F_2(\theta_2^{(t,k)}), \theta_1^{(t,k)} - \theta_2^{(t,k)} \right\rangle \\
&\leq -\left\langle \nabla F(\theta_1^{(t,k)}) - \nabla F(\theta_2^{(t,k)}), \theta_1^{(t,k)} - \theta_2^{(t,k)} \right\rangle + 2\Omega\|\theta_1^{(t,k)} - \theta_2^{(t,k)}\| \\
&\leq -\frac{1}{L}\|\nabla F(\theta_1^{(t,k)}) - \nabla F(\theta_2^{(t,k)})\|^2 + 2\Omega\|\theta_1^{(t,k)} - \theta_2^{(t,k)}\| \quad \text{(by smoothness and convexity)} \\
&\leq -\frac{1}{L}\|\nabla F(\theta_1^{(t,k)}) - \nabla F(\theta_2^{(t,k)})\|^2 + \frac{1}{6\eta K}\|\theta_1^{(t,k)} - \theta_2^{(t,k)}\|^2 + 6\eta K\Omega^2 \quad \text{(by Young's inequality)}
\end{aligned}
$$

The last term is bounded as follows

$$
\begin{aligned}
&\|g_1(\theta_1^{(t,k)}) - g_2(\theta_2^{(t,k)})\|^2 \\
&\leq 5\Big(\|g_1(\theta_1^{(t,k)}) - \nabla F_1(\theta_1^{(t,k)})\|^2 + \|\nabla F_1(\theta_1^{(t,k)}) - \nabla F(\theta_1^{(t,k)})\|^2 + \|\nabla F(\theta_1^{(t,k)}) - \nabla F(\theta_2^{(t,k)})\|^2 \\
&\quad + \|\nabla F(\theta_2^{(t,k)}) - \nabla F_2(\theta_2^{(t,k)})\|^2 + \|g_2(\theta_2^{(t,k)}) - \nabla F_2(\theta_2^{(t,k)})\|^2\Big) \\
&\leq 5\|\nabla F(\theta_1^{(t,k)}) - \nabla F(\theta_2^{(t,k)})\|^2 + 10(\widehat{\sigma}^2 + \Omega^2) \quad \text{(by (15) and Assumption 5)}
\end{aligned}
$$

Substituting the above four bounds back to (20) gives (note that $\eta \leq \frac{1}{3L}$)

$$
\begin{aligned}
\mathbf{E}\Big[\|\theta_1^{(t,k+1)} - \theta_2^{(t,k+1)}\|^2\Big] &\leq \Big(1 + \frac{1}{K}\Big)\|\theta_1^{(t,k)} - \theta_2^{(t,k)}\|^2 - \eta\Big(\frac{2}{L} - 5\eta\Big)\|\nabla F(\theta_1^{(t,k)}) - \nabla F(\theta_2^{(t,k)})\|^2 \\
&\quad + 6\eta^2 K\widehat{\sigma}^2 + 12\eta^2 K\Omega^2 + 10\eta^2(\widehat{\sigma}^2 + \Omega^2) \\
&\leq \Big(1 + \frac{1}{K}\Big)\|\theta_1^{(t,k)} - \theta_2^{(t,k)}\|^2 + 6\eta^2 K\widehat{\sigma}^2 + 12\eta^2 K\Omega^2 + 10\eta^2(\widehat{\sigma}^2 + \Omega^2).
\end{aligned}
$$

Unrolling recursively, we obtain

$$
\begin{aligned}
\mathbf{E}\Big[\|\theta_1^{(t,k+1)} - \theta_2^{(t,k+1)}\|^2\Big] &\leq \frac{(1 + 1/K)^K - 1}{1/K}\Big[6\eta^2 K\widehat{\sigma}^2 + 12\eta^2 K\Omega^2 + 10\eta^2(\widehat{\sigma}^2 + \Omega^2)\Big] \\
&\leq 12\eta^2 K^2\widehat{\sigma}^2 + 24\eta^2 K^2\Omega^2 + 20\eta^2 K(\widehat{\sigma}^2 + \Omega^2) \\
&\leq \eta^2(24K^2\bar{\Omega}^2 + 20K\bar{\Omega}^2).
\end{aligned}
$$

where we use the fact that $\frac{(1 + 1/K)^K - 1}{1/K} \leq K(e - 1) \leq 2K$, and $\bar{\Omega}^2 := \max\{\widehat{\sigma}^2, \Omega^2\}$.

By convexity, for any $i$,

$$
\mathbf{E}\Big[\|\theta_i^{(t,k+1)} - \bar{\theta}^{(t,k+1)}\|^2\Big] \leq \eta^2(24K^2\bar{\Omega}^2 + 20K\bar{\Omega}^2).
$$

$$\square$$

Substituting the result of Lemma 4 to Lemma 3, and telescoping over $t$, we obtain

$$
\mathbb{E}\Big[\frac{1}{T}\sum_{t=0}^{T-1}\frac{1}{K}\sum_{k=0}^{K-1} F(\bar{\theta}^{(t,k)}) - F(\theta^*)\Big] \leq \frac{D^2}{2\eta KT} + 2\eta\widehat{\sigma}^2\Lambda + \eta^2 L(24K^2\bar{\Omega}^2 + 20K\bar{\Omega}^2),
$$

where $D := \|\theta^{(0)} - \theta^*\|$, $\Lambda := \sum_{i=1}^{m} \lambda_i^2$. By optimizing $\eta$ on the R.H.S, we obtain

$$\mathbb{E}\Big[\frac{1}{KT}\sum_{t=0}^{T-1}\sum_{k=0}^{K-1} F(\bar{\theta}^{(t,k)}) - F(\theta^*)\Big] \leq \mathcal{O}\Big(\frac{LD^2}{KT} + \frac{\widehat{\sigma}D\Lambda^{\frac{1}{2}}}{\sqrt{KT}} + \frac{L^{\frac{1}{3}}\bar{\Omega}^{\frac{2}{3}}D^{\frac{4}{3}}}{K^{\frac{1}{3}}T^{\frac{2}{3}}} + \frac{L^{\frac{1}{3}}\bar{\Omega}^{\frac{2}{3}}D^{\frac{4}{3}}}{T^{\frac{2}{3}}}\Big),$$

when

$$\eta = \min\left\{\frac{1}{3L}, \frac{D}{2\sqrt{KT\Lambda}\widehat{\sigma}}, \frac{D^{\frac{2}{3}}}{48^{\frac{1}{3}}K^{\frac{2}{3}}T^{\frac{1}{3}}L^{\frac{1}{3}}\bar{\Omega}^{\frac{2}{3}}}, \frac{D^{\frac{2}{3}}}{40^{\frac{1}{3}}KT^{\frac{1}{3}}L^{\frac{1}{3}}\bar{\Omega}^{\frac{2}{3}}}\right\}.$$

## D  PROOF OF LEMMA 1

We first prove the following fact:

**Fact 1:**

$$(a) \quad \mathcal{L}(Q, f) \leq \mathcal{L}_\rho^\gamma(P_\lambda, f), \qquad \forall f \in \mathcal{F}, Q \in \mathcal{B}(P_\lambda, \rho).$$
$$(b) \quad \inf_{f' \in \mathcal{F}} \mathcal{L}(Q, f') \leq \inf_{f' \in \mathcal{F}} \mathcal{L}_\rho^\gamma(P_\lambda, f'), \quad \forall Q \in \mathcal{B}(P_\lambda, \rho).$$

For (a), we have

$$\mathcal{L}(Q, f) \leq \sup_{P' \in \mathcal{B}(P_\lambda, \rho)} \mathcal{L}(P', f) = \inf_{\gamma' \geq 0}\Big\{\gamma'\rho^2 + \mathbf{E}_{Z \sim P_\lambda}\Big[\phi_\gamma(Z, f)\Big]\Big\}$$

$$\leq \gamma\rho^2 + \mathbf{E}_{Z \sim P_\lambda}\Big[\phi_\gamma(Z, f)\Big] =: \mathcal{L}_\rho^\gamma(P_\lambda, f),$$

where the equality is due to strong duality result by Gao & Kleywegt (2016).

For (b), defining $f_{P_\lambda} := \arg\min_{f' \in \mathcal{F}} \mathcal{L}_\rho^\gamma(P_\lambda, f')$, we have

$$\inf_{f' \in \mathcal{F}} \mathcal{L}(Q, f') \leq \mathcal{L}(Q, f_{P_\lambda}) \leq \sup_{P' \in \mathcal{B}(P_\lambda, \rho)} \mathcal{L}(P', f_{P_\lambda}) \tag{21}$$

$$= \inf_{\gamma' \geq 0}\Big\{\gamma'\rho^2 + \mathbf{E}_{Z \sim P_\lambda}\Big[\phi_\gamma(Z, f_{P_\lambda})\Big]\Big\} \tag{22}$$

$$\leq \gamma\rho^2 + \mathbf{E}_{Z \sim P_\lambda}\Big[\phi_\gamma(Z, f_{P_\lambda})\Big] \tag{23}$$

$$= \inf_{f' \in \mathcal{F}} \mathcal{L}_\rho^\gamma(P_\lambda, f'). \tag{24}$$

We next prove the second fact:

**Fact 2:**

$$(a) \quad \mathcal{L}_\rho^\gamma(P_\lambda, f) \leq \mathcal{L}(Q, f) + 2L_z\rho + |\gamma - \gamma^*|\rho^2, \quad \forall f \in \mathcal{F}, Q \in \mathcal{B}(P_\lambda, \rho)$$
$$(b) \quad \inf_{f' \in \mathcal{F}} \mathcal{L}_\rho^\gamma(P_\lambda, f') \leq \inf_{f' \in \mathcal{F}} \mathcal{L}(Q, f') + 2L_z\rho + |\gamma - \gamma^*|\rho^2.$$

For (a), we have:

$$\mathcal{L}_\rho^\gamma(P_\lambda, f) = \Big\{\sup_{P' \in \mathcal{B}(P_\lambda, \rho)} \mathcal{L}(P', f)\Big\} + \Big\{\mathcal{L}_\rho^\gamma(P_\lambda, f) - \sup_{P' \in \mathcal{B}(P_\lambda, \rho)} \mathcal{L}(P', f)\Big\}$$

$$\leq \Big\{\mathcal{L}(Q, f) + 2L_z\rho\Big\} + \Big\{\mathbf{E}_{Z \sim P_\lambda}[\phi_\gamma(Z, f)] + \rho^2\gamma - \min_{\gamma' \geq 0}\big\{\rho^2\gamma' + \mathbf{E}_{Z \sim P_\lambda}[\phi_{\gamma'}(Z, f)]\big\}\Big\}$$

$$\leq \mathcal{L}(Q, f) + 2L_z\rho + \rho^2(\gamma - \gamma^*) + \mathbf{E}_{Z \sim P}\Big[\phi_\gamma(Z, f) - \phi_{\gamma^*}(Z, f)\Big]$$

$$= \mathcal{L}(Q, f) + 2L_z\rho + \rho^2(\gamma - \gamma^*) + \mathbf{E}_{Z \sim P}\Big[\sup_{\zeta \in \mathcal{Z}}\big\{\ell(\zeta, h) - \gamma d(\zeta, Z)\big\} - \sup_{\zeta \in \mathcal{Z}}\big\{\ell(\zeta, h) - \gamma^* d^2(\zeta, Z)\big\}\Big]$$

$$= \mathcal{L}(Q, f) + 2L_z\rho + (\gamma - \gamma^*)\Big(\rho^2 - \mathbf{E}_{Z \sim P}\Big[\sup_{\zeta \in \mathcal{Z}} d^2(\zeta, Z)\Big]\Big)$$

$$\leq \mathcal{L}(Q, f) + 2L_z\rho + |\gamma - \gamma^*|\rho^2,$$

where the first inequality is due to Proposition 1, and the last inequality is because we choose $\gamma \geq L_z/\rho$ and that fact that $\gamma^* \leq L_z/\rho$ by Lemma 1 of Lee & Raginsky (2018).

For (b), defining $f_Q := \arg\min_{f \in \mathcal{F}} \mathcal{L}(Q, f)$, we have

$$\inf_{f' \in \mathcal{F}} \mathcal{L}_\rho^\gamma(P_\lambda, f') \leq \mathcal{L}_\rho^\gamma(P_\lambda, f_Q) \tag{25}$$

$$\leq \mathcal{L}(Q, f_Q) + 2L_z\rho + |\gamma - \gamma^*|\rho^2 \tag{26}$$

$$= \inf_{f' \in \mathcal{F}} \mathcal{L}(Q, f') + 2L_z\rho + |\gamma - \gamma^*|\rho^2, \tag{27}$$

where the second line is due to **Fact 2**(a).

Combining all facts, we complete the proof. Specifically, by adding two inequalities in **Fact 1**(a) and **Fact 2**(b), we obtain the upperbound of Lemma 1. Similarly, adding two inequalities in **Fact 1**(b) and **Fact 2**(a), we obtain the lowerbound of this lemma.

Finally, we provide the proof of the following proposition that was used in proving **Fact 2**(a).

**Proposition 1.** *Let Assumption 2 (a) holds. For any $f \in \mathcal{F}$ and for all $Q \in \mathcal{B}(P_\lambda, \rho)$, we have*

$$\sup_{P' \in \mathcal{B}(P_\lambda, \rho)} \mathcal{L}(P', f) \leq \mathcal{L}(Q, f) + 2L_z\rho.$$

*Proof.* Denote $P^* := \arg\max_{P' \in \mathcal{B}(P_\lambda, \rho)} \mathcal{L}(P', f)$. We have

$$\sup_{P' \in \mathcal{B}(P_\lambda, \rho)} \mathcal{L}(P', f) = \mathcal{L}(Q, f) + \sup_{P' \in \mathcal{B}(P_\lambda, \rho)} \mathcal{L}(P', f) - \mathcal{L}(Q, f)$$
$$\leq \mathcal{L}(Q, f) + |\mathcal{L}(P^*, f) - \mathcal{L}(Q, f)|,$$
$$\leq \mathcal{L}(Q, f) + L_z|\mathbf{E}_{Z \sim P^*}[\ell(Z, h)/L_z] - \mathbf{E}_{Z \sim Q}[\ell(Z, h)/L_z]|$$
$$\leq \mathcal{L}(Q, f) + L_z W_1(P^*, Q)$$
$$\leq \mathcal{L}(Q, f) + L_z[W_2(P^*, P_\lambda) + W_2(P_\lambda, Q)] \tag{28}$$
$$\leq \mathcal{L}(Q, f) + L_z 2\rho,$$

where the fourth line are due to Kantorovich-Rubinstein dual representation theorem, i.e.,

$$W_1(P, Q) = \sup_h \left\{ \mathbf{E}_{Z \sim P}[h(Z)] - \mathbf{E}_{Z \sim Q}[h(Z)] : h(\cdot) \text{ is 1-Lipschitz} \right\}$$

and the fifth line is due to $W_1(P^*, Q) \leq W_2(P^*, Q)$ and triangle inequality. $\square$

# E  PROOF OF THEOREM 2

*Proof.* To simplify notation, we denote $\Phi := \phi_\gamma \circ \mathcal{F} = \{z \mapsto \phi_\gamma(z, f), f \in \mathcal{F}\}$ where $\mathcal{F} = \{f_\theta, \theta \in \Theta \subset \mathbb{R}^d\}$, which represents the composition of $\phi_\gamma$ with each of the loss function $f_\theta$ parametrized by $\theta$ belonging to the parameter class $\Theta$.

Defining $f_{P_\lambda} \in \arg\min_{f \in \mathcal{F}} \mathcal{L}_\rho^\gamma(P_\lambda, f)$ and $\widehat{\theta}^* \in \underset{\theta \in \Theta}{\arg\min} \ \mathbf{E}_{Z \sim \widehat{P}_\lambda}[\phi_\gamma(Z, f_\theta)]$ such that $\mathcal{L}_\rho^\gamma(\widehat{P}_\lambda, f_{\theta^*}) = \inf_{\theta \in \Theta}\left[\mathbf{E}_{Z \sim \widehat{P}_\lambda}[\phi_\gamma(Z, f_\theta)] + \gamma\rho^2\right]$, we decompose the excess risk as follows:

$$\mathcal{E}_\rho^\gamma(P_\lambda, f_{\widehat{\theta}^\varepsilon}) = \mathcal{L}_\rho^\gamma(P_\lambda, f_{\widehat{\theta}^\varepsilon}) - \inf_{f \in \mathcal{F}} \mathcal{L}_\rho^\gamma(P_\lambda, f)$$

$$= \mathcal{L}_\rho^\gamma(P_\lambda, f_{\widehat{\theta}^\varepsilon}) - \mathcal{L}_\rho^\gamma(P_\lambda, f_{P_\lambda})$$

$$= \left[ \mathcal{L}_\rho^\gamma(P_\lambda, f_{\widehat{\theta}^\varepsilon}) - \mathcal{L}_\rho^\gamma(\widehat{P}_\lambda, f_{\widehat{\theta}^\varepsilon}) \right] + \underbrace{\left[ \mathcal{L}_\rho^\gamma(\widehat{P}_\lambda, f_{\widehat{\theta}^\varepsilon}) - \mathcal{L}_\rho^\gamma(\widehat{P}_\lambda, f_{\widehat{\theta}^*}) \right]}_{\leq \varepsilon}$$

$$+ \underbrace{\left[ \mathcal{L}_\rho^\gamma(\widehat{P}_\lambda, f_{\widehat{\theta}^*}) - \mathcal{L}_\rho^\gamma(\widehat{P}_\lambda, f_{P_\lambda}) \right]}_{\leq 0} + \left[ \mathcal{L}_\rho^\gamma(\widehat{P}_\lambda, f_{P_\lambda}) - \mathcal{L}_\rho^\gamma(P_\lambda, f_{P_\lambda}) \right]$$

$$\leq 2 \sup_{\phi_\gamma \in \Phi} \left| \mathbf{E}_{Z \sim P_\lambda}[\phi_\gamma(Z, f_\theta)] - \mathbf{E}_{Z \sim \widehat{P}_\lambda}[\phi_\gamma(Z, f_\theta)] \right| + \varepsilon$$

$$\leq 2 \sup_{\phi_\gamma \in \Phi} \sum_{i=1}^m \lambda_i \left| \mathbf{E}_{Z_i \sim P_i}[\phi_\gamma(Z_i, f_\theta)] - \mathbf{E}_{Z_i \sim \widehat{P}_i}[\phi_\gamma(Z_i, f_\theta)] \right| + \varepsilon$$

$$\leq 2 \sum_{i=1}^m \lambda_i \sup_{\phi_\gamma \in \Phi} \left| \mathbf{E}_{Z_i \sim P_i}[\phi_\gamma(Z_i, f_\theta)] - \mathbf{E}_{Z_i \sim \widehat{P}_i}[\phi_\gamma(Z_i, f_\theta)] \right| + \varepsilon$$

$$\leq \sum_{i=1}^m \lambda_i \left[ 4\mathcal{R}_i(\Phi) + 2M_\ell \sqrt{\frac{2\log(2m/\delta)}{n_i}} \right] + \varepsilon \text{ with probability at least } 1 - \delta, \quad (29)$$

where the first inequality is due to optimization error and definition of $\widehat{\theta}^*$. The second inequality is due to the fact that $|\sum_{i=1}^m \lambda_i a_i| \leq \sum_{i=1}^m \lambda_i |a_i|, \forall a_i \in \mathbb{R}$ and $\lambda_i \geq 0$. The third inequality is because pushing the sup inside increases the value. For the last inequality, using the facts that (i) $|\phi_\gamma(z, f)| \leq M_\ell$ due to $-M_\ell \leq \ell(z, h) \leq \phi_\gamma(z, f) \leq \sup_{z \in \mathcal{Z}} \ell(z, h) \leq M_\ell$ and (ii) the Rademacher complexity of the function class $\Phi$ defined by $\mathcal{R}_i(\Phi) = \mathbf{E}[\sup_{\phi_\gamma \in \Phi} \frac{1}{n_i} \sum_{k=1}^{n_i} \sigma_k \phi_\gamma(Z_k, f_\theta)]$ where the expectation is w.r.t both $Z_k \overset{\text{i.i.d.}}{\sim} P_i$ and i.i.d. Rademacher random variable $\sigma_k$ independent of $Z_k, \forall k \in [n_i]$, we have

$$\sup_{\phi_\gamma \in \Phi} \left| \mathbf{E}_{Z_i \sim P_i}[\phi_\gamma(Z_i, f_\theta)] - \mathbf{E}_{Z_i \sim \widehat{P}_i}[\phi_\gamma(Z_i, f_\theta)] \right| \geq 2\mathcal{R}_i(\Phi) + M_\ell \sqrt{\frac{2\log(2m/\delta)}{n_i}} \quad (30)$$

with probability $\leq \delta/m$ due to the standard symmetrization argument and McDiarmid's inequality (Shalev-Shwartz & Ben-David, 2014, Theorem 26.5). Multiplying $\lambda_i$ to both sides of (30), summing up the inequalities over all $i \in [n]$, and using union bound, we obtain (29).

Define a stochastic process $\left( X_{\phi_\gamma} \right)_{\phi_\gamma \in \Phi}$

$$X_{\phi_\gamma} := \frac{1}{\sqrt{n_i}} \sum_{k=1}^{n_i} \sigma_k \phi_\gamma(Z_k, f_\theta)$$

which is zero-mean because $\mathbf{E}\left[ X_{\phi_\gamma} \right] = 0$ for all $\phi_\gamma \in \Phi$. To upper-bound $\mathcal{R}_n(\Phi)$, we first show that $\left( X_{\phi_\gamma} \right)_{\phi_\gamma \in \Phi}$ is a sub-Gaussian process with respect to the following pseudometric

$$\left\| \phi_\gamma - \phi_\gamma' \right\|_\infty := \sup_{z \in \mathcal{Z}} \left| \phi_\gamma(z, f_\theta) - \phi_\gamma(z, f_{\theta'}) \right|. \quad (31)$$

For any $t \in \mathbb{R}$, using Hoeffding inequality with the fact that $\sigma_k, k \in [n]$, are i.i.d. bounded random variable with sub-Gaussian parameter 1, we have

$$\mathbf{E}\left[ \exp\left( t\left( X_{\phi_\gamma} - X_{\phi_\gamma'} \right) \right) \right] = \mathbf{E}\left[ \exp\left( \frac{t}{\sqrt{n_i}} \sum_{k=1}^{n_i} \sigma_k \left( \phi_\gamma(Z_k, f_\theta) - \phi(Z_k, f_{\theta'}) \right) \right) \right]$$

$$= \left( \mathbf{E}\left[ \exp\left( \frac{t}{\sqrt{n_i}} \sigma_1 \left( \phi_\gamma(Z_1, f_\theta) - \phi_\gamma(Z_1, f_{\theta'}) \right) \right) \right] \right)^{n_i}$$

$$\leq \exp\left( \frac{t^2 \left\| \phi_\gamma - \phi_\gamma' \right\|_\infty^2}{2} \right).$$

Then, invoking Dudley entropy integral, we have

$$\sqrt{n_i}\,\mathcal{R}_i(\Phi) = \mathbf{E} \sup_{\phi_\gamma \in \Phi} X_{\phi_\gamma} \leq 12 \int_0^\infty \sqrt{\log \mathcal{N}(\Phi, \|\cdot\|_\infty, \epsilon)} \mathrm{d}\epsilon \tag{32}$$

We will show that when $\theta \mapsto \ell(z, h_\theta)$ is $L_\theta$-Lipschitz by Assumption 2, then $\theta \mapsto \phi_\gamma(z, f_\theta)$ is also $L_\theta$-Lipschitz as follows.

$$
\begin{aligned}
\left| \phi_\gamma(z, f_\theta) - \phi_\gamma(z, f_{\theta'}) \right| &= \left| \sup_{\zeta \in \mathcal{Z}} \inf_{\zeta' \in \mathcal{Z}} \left\{ \ell(\zeta, h_\theta) - \gamma d(\zeta, z) - \ell(\zeta', h_{\theta'}) + \gamma d(\zeta', z) \right\} \right| \\
&\leq \left| \sup_{\zeta \in \mathcal{Z}} \left\{ \ell(\zeta, h_\theta) - \ell(\zeta, h_{\theta'}) \right\} \right| \\
&\leq \sup_{\zeta \in \mathcal{Z}} \left| \ell(\zeta, h_\theta) - \ell(\zeta, h_{\theta'}) \right| \\
&\leq L_\theta \|\theta - \theta'\|,
\end{aligned}
$$

which implies

$$\left\| \phi_\gamma - \phi'_\gamma \right\|_\infty \leq L_\theta \|\theta - \theta'\|.$$

Therefore, by contraction principle (Shalev-Shwartz & Ben-David, 2014), we have

$$\mathcal{N}(\Phi, \|\cdot\|_\infty, \epsilon) \leq \mathcal{N}(\Theta, \|\cdot\|, \epsilon/L_\theta). \tag{33}$$

Substituting (33) and (32) into (29), we obtain

$$\mathcal{E}_\rho^\gamma(P_\lambda, f_{\widehat{\theta^\varepsilon}}) \leq \sum_{i=1}^m \lambda_i \left[ \frac{48\mathcal{C}(\Theta)}{\sqrt{n_i}} + 2M_\ell \sqrt{\frac{2\log(2m/\delta)}{n_i}} \right] + \varepsilon, \tag{34}$$

which will be substituted into the upper-bound in Lemma 1 to complete the proof. $\qquad \square$

## F    PROOF OF CORROLARY 1

We now present how we adapt the result from Fournier & Guillin (2015) to prove Corollary 1

**Proposition 2** (Measure concentration (Fournier & Guillin, 2015, Theorem 2)). *Let $P$ be a probability distribution on a bounded set $\mathcal{Z}$. Let $\widehat{P}_n$ denote the empirical distribution of $Z_1, \ldots, Z_n \overset{i.i.d.}{\sim} P$. Assuming that there exist constants $a > 1$ such that $A := \mathbf{E}_{Z \sim P}\left[\exp(\|Z\|^a)\right] < \infty$ (i.e., $P$ is a light-tail distribution). Then, for any $\rho > 0$,*

$$\mathbf{P}\left[ W_p(\widehat{P}_n, P) \geq \rho \right] \leq \begin{cases} c_1 \exp\left(-c_2 n \rho^{\max\{d/p, 2\}}\right) & \text{if } \rho \leq 1 \\ c_1 \exp\left(-c_2 n \rho^a\right) & \text{if } \rho > 1 \end{cases}$$

*where $c_1, c_2$ are constants depending on $a$, $A$ and $d$.*

As a consequence of this proposition, for any $\delta > 0$, we have

$$\mathbf{P}\left[ W_2(\widehat{P}_n, P) \leq \widehat{\rho}_n^\delta \right] \geq 1 - \delta \quad \text{where} \quad \widehat{\rho}_n^\delta := \begin{cases} \left( \frac{\log(c_1/\delta)}{c_2 n} \right)^{\min\{2/d, 1/2\}} & \text{if } n \geq \frac{\log(c_1/\delta)}{c_2}, \\ \left( \frac{\log(c_1/\delta)}{c_2 n} \right)^{1/\alpha} & \text{if } n < \frac{\log(c_1/\delta)}{c_2}. \end{cases} \tag{35}$$

In Proposition 2, Fournier & Guillin (2015) show that the empirical distribution $\widehat{P}_n$ converges in Wasserstein distance to the true $P$ at a specific rate. This implies that judiciously scaling the radius of Wasserstein balls according to (35) provides natural confidence regions for the data-generating distribution $P$.

By the duality of transport cost (Santambrogio, 2015, p.261), we have

$$W_p^p(\mu, \nu) = \sup_{\varphi(x) + \psi(y) \leq d^p(x,y)} \int \varphi \, d\mu + \psi \, d\nu = \sup_{\varphi(x) + \psi(y) \leq d^p(x,y)} T_f(\mu, \nu), \quad \forall p \geq 1,$$

which is the supremum of linear functionals $T_f : \mathcal{P} \times \mathcal{P} \mapsto \mathbb{R}$ defined by $T_f(\mu, \nu) = \langle(\mu, \nu), (\varphi, \psi)\rangle$; therefore, $(\mu, \nu) \mapsto W_p^p(\mu, \nu)$ is convex. Thus we have

$$W_2^2(P_\lambda, \widehat{P}_\lambda) = W_2^2\Big(\sum_{i=1}^m \lambda_i(\widehat{P}_{n_i}, P_i)\Big) \leq \sum_{i=1}^m \lambda_i W_2^2(\widehat{P}_{n_i}, P_i). \tag{36}$$

Then, we have

$$\begin{aligned}
\mathbf{P}\Big[W_2(P_\lambda, \widehat{P}_\lambda) \geq \sqrt{\sum_{i=1}^m \lambda_i \widehat{\rho}_{n_i}^{\delta/m}}\Big] &= \mathbf{P}\Big[W_2^2(P_\lambda, \widehat{P}_\lambda) \geq \sum_{i=1}^m \lambda_i \widehat{\rho}_{n_i}^{\delta/m}\Big] \\
&\leq \mathbf{P}\Big[\sum_{i=1}^m \lambda_i W_2^2(\widehat{P}_{n_i}, P_i) \geq \sum_{i=1}^m \lambda_i \widehat{\rho}_{n_i}^{\delta/m}\Big] \\
&\leq \sum_{i=1}^m \mathbf{P}\Big[W_2^2(\widehat{P}_{n_i}, P_i) \geq \widehat{\rho}_{n_i}^{\delta/m}\Big] \\
&= \sum_{i=1}^m \mathbf{P}\Big[W_2(\widehat{P}_{n_i}, P_i) \geq \widehat{\rho}_{n_i}^{\delta/2m}\Big] \\
&\leq \sum_{i=1}^m \frac{\delta}{2m} = \frac{\delta}{2},
\end{aligned} \tag{37}$$

where the first inequality is due to (36), the second inequality is due to the union bound, and the last inequality is due to Proposition 2 and (35).

According to (28), by setting $\rho = \Big(\sum_{i=1}^m \lambda_i \widehat{\rho}_{n_i}^{\delta/m}\Big)^{1/2}$ in Theorem 2 and using union bound, we complete the proof.

## G  ADDITIONAL EXPERIMENTAL SETTINGS AND RESULTS

### G.1  DATASETS

Table 1: Statistics of all datasets using in the experiments.

| Dataset | $m$ | Total samples | Num labels / client | Samples / client Mean | Std |
|---|---|---|---|---|---|
| CIFAR-10 | 20 | 43,098 | 3 | 2154 | 593.8 |
| MNIST | 100 | 70,000 | 2 | 700 | 313.4 |
| MNIST-M | 100 | 70,000 | 2 | 700 | 322.4 |

We distribute all datasets to clients as follows:

- **MNIST**: A handwritten digit dataset (Lecun et al., 1998) including $70,000$ instances belonged to 10 classes. We distribute dataset to $m = 100$ clients and each client has a different local data size with only 2 of the 10 classes.
- **CIFAR-10**: An object recognition dataset (Krizhevsky, 2009) including $60,000$ colored images belonged to 10 classes. We partition the dataset to $m = 20$ clients and there are 3 labels per client. Each client has a different local data size.
- **Three handwritten digit**: MNIST, MNIST-M (Ganin & Lempitsky, 2015), and USPS (Hull, 1994). We distribute one dataset, for example, MNIST to 100 source clients, and leave MNIST-M to the target client. Each source client has only 2 labels over 10 labels. As the number of data samples in USPS is relatively small amount (9,298 samples), we only use this dataset for the target client.

We standardize and randomly split all datasets with $75\%$ and $25\%$ for training and testing, respectively. In domain adaptation, the target client only has test data. The statistics of all datasets are summarized in Tab. 1.

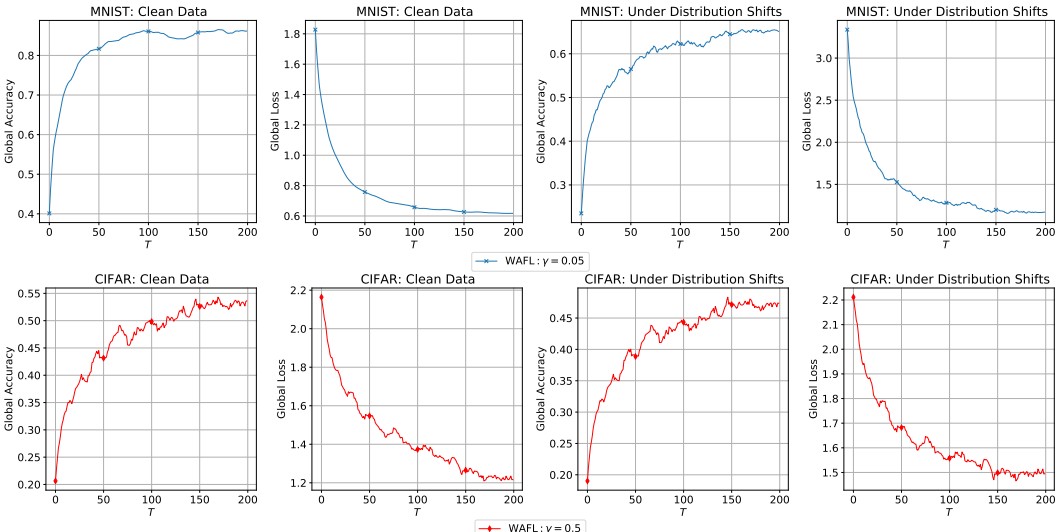

Figure 5: Convergence of WAFL.

## G.2 MODELS

The details of models for each dataset is provided as follows:

- **MNIST**: We use a multinomial logistic regression model (MLR) with a cross-entropy loss function and an $L_2$-regularization term.
- **CIFAR-10**: We use a CNN model employed in McMahan et al. (2017).
- **Set of three handwriten digits**: We use a multinomial logistic regression model (MLR) with a cross-entropy loss function and an $L_2$-regularization term.

In all settings, we randomly sample 10 clients to participate in training the global robust model in each communication round, and set the number of local epochs to $K = 2$ and the number of communication rounds to $T = 200$. All experiments were conducted using PyTorch (Paszke et al., 2019).

## G.3 CONVERGENCE OF WAFL

We verify the convergence of WAFL under two cases: *clean data* (no attacked clients) and *distribution shifts* (where $40\%$ of clients are attacked). In each case, we use two datasets: MNIST and CIFAR-10 and employ the same setup as in Sec. 6. Specifically, for MNIST, we distribute the dataset to 100 clients and set $\gamma = 0.05$. For CIFAR-10, we use 20 clients and set $\gamma = 0.5$. We use $T = 200$ communication iterations.

To show WAFL's convergence, we plot both the *original* loss (using the function $\ell$) and global accuracy in Fig. 5.

## G.4 DOMAIN ADAPTATION.

We show that WAFL demonstrates its superior capability in domain adaptation by collaboratively learning from a multi-source domain $\widehat{P}_\lambda$, and applying it to an unseen, but related, target domain $\widehat{Q}$. In federated domain adaptation (Peng et al., 2019; Liu et al., 2021), source domains often need access to the private data of target domains to find a common representation to aid training. However, this violates the data privacy assumption of FL. By contrast, the construction of the model in WAFL does not involve such private data.

Our experimental set up is as follows. We consider three datasets: MNIST (*mt*), MNIST-M (*mm*) and USPS (*up*), which are widely used in domain adaptation. Among these datasets, we consider

Table 2: Domain adaptation performance of WAFL and FedAvg as an accuracy on the target dataset.

| Algorithm | $mt{\rightarrow}mm$ | $mt{\rightarrow}up$ | $mm{\rightarrow}up$ | $mm{\rightarrow}mt$ | Average |
|---|---|---|---|---|---|
| WAFL | **40.23** | **66.94** | **62.04** | **79.27** | **62.12** |
| FedAvg | 37.71 | 66.08 | 58.60 | 75.80 | 59.55 |

one dataset as a source domain (distributed to 100 clients), and another dataset as a target domain (assumed to be on one client). For example, if $mt$ is used as the source and $mm$ the domain, we use $mt{\rightarrow}mm$ to denote this case. Specifically, the $mm$ dataset is distributed to 100 clients to learn a global model, then the model is tested on the $mt$ dataset. We compare WAFL against the vanilla FedAvg as a baseline, and report the accuracy on the target dataset in several scenarios in Tab. 2.

WAFL markedly improves from FedAvg in all cases. On average, the accuracy of WAFL is 2.5 percentage point higher than that of FedAvg. This indicates WAFL's benefits in transferring knowledge from multi-sourced, private client distributions to a new target distribution.

