# OpenReview forum: "ON THE GENERALIZATION OF WASSERSTEIN ROBUST FEDERATED LEARNING"
_ICLR.cc/2022/Conference — ICLR 2022 Submitted_

### Official Review · Reviewer_1hSQ · 2021-10-17

**Correctness:** 3
**Technical Novelty And Significance:** 3
**Empirical Novelty And Significance:** 2
**Recommendation:** 5
**Confidence:** 4

**Main Review:**

**Major commments:**

- The theoretical part of WAFL is systematic and well-written, with analysis from both optimization and statistics aspects.

- My biggest concern lies in the motivation/marginal contribution of WAFL against existing agnostic FL framework such as (Mohri et al., 2019); (Deng et al., 2020b). On page 4, the paper compares WAFL with counterparts in terms of uncertainty set, and conclude that Wasserstein ball is able to cover the uncertainty set of agnostic FL. However, a larger uncertainty ball may not be an advantage, it will also leads to over-conservative decisions. Therefore, a crucial question for the authors is: for what types of application can WAFL outperform existing method? If so, is there any theoretical or empirical evidence?

- The paper didn't report a systematic way to choose $\gamma$ in the numerical experiment part, the choices of $\gamma = 0.5$ for MNIST and $\gamma = 10$ for CIFAR-10 are not well-supported.

- For the numerical part, it would be better to include (Mohri et al., 2019) or (Deng et al., 2020b) as benchmark as well. This would be helpful to clarify my first point.

**Minor Comments:**

- It is claimed on page 3 that ``In FL, some existing works have explored the Wasserstein distance to enhance robustness (Rei- sizadeh et al., 2020; Diamandis et al., 2021; Du et al., 2020; Deng et al., 2020b).`` Though Du et al., 2020 and Deng et al., 2020b are examples of distributional robust FL, I can not find their connection to Wasserstein distance. Did I miss something here?

**Summary Of The Paper:**

The paper proposes a Wasserstein-based distributionally robust federated learning (WAFL) framework to address the statistical heterogeneity problem, which is also applicable to distributional shift setting and domain adaptation. Inspired by (Sinha et al., 2020), the WAFL problem is reformulated into the dual form, and the Lagrangian ($\gamma$) is chosen as a fixed value reflecting the algorithm's conservatives. The paper develops a local SGD scheme to solve the problem and provides a theoretical convergence guarantee under a set of standard assumptions. In addition, the paper also provides a generalization bound for the proposed algorithm by resorting to standard techniques such as covering number, entropy integral, and empirical convergence rate of Wasserstein distances. Finally, the results of several numerical experiments are reported to justify the performance of the algorithm.

**Summary Of The Review:**

In general, the paper is well-written. It provides a comprehensive study of the proposed framework. However, in my humble view, the motivation of the paper is not well-supported, and the comparison against existing methods is insufficient. Therefore, I rate this paper as a marginal paper.

---

> ### Author Response · Authors · 2021-11-19
> **Response to Reviewer 1hSQ**
>
> > My biggest concern lies in the motivation/marginal contribution of WAFL against existing agnostic FL frameworks such as (Mohri et al., 2019); (Deng et al., 2020b). On page 4, the paper compares WAFL with counterparts in terms of uncertainty set, and concludes that Wasserstein ball is able to cover the uncertainty set of agnostic FL. However, a larger uncertainty ball may not be an advantage, it will also lead to over-conservative decisions. Therefore, a crucial question for the authors is: for what types of application can WAFL outperform existing methods? If so, is there any theoretical or empirical evidence?
>
> Thank you for raising this point. We would like to clarify that the main purpose of the comparison between WAFL and counterparts, such as Agnostic FL (Mohri et al., 2019, Deng et al. 2020b) and Adversarial FL (Reisizadeh et al., 2020), is to demonstrate the flexibility of the Wasserstein ball in covering other ambiguity sets. We agree with the reviewer that simply enlarging an ambiguity set does not imply a more desirable classifier. The point we are making is that by appropriately setting the values of $\rho$ and $\lambda$, the Wasserstein ball can cover the ambiguity sets defined in Agnostic FL and Adversarial FL. This ball need not to be significantly “larger.”
>
> **Compared with agnostic FL, the main advantage of WAFL is flexibility.** The ambiguity set of agnostic FL is simply the convex hull of the clients’ distributions. With appropriately chosen $\rho$ and $\lambda$, WAFL can create a Wasserstein ball enclosing this convex hull. WAFL also gives the designer flexibility in choosing $\lambda$ to make the ball larger or smaller, therefore not confined to the hull.
>
> **Compared with adversarial robust FL, the key advantage of WAFL is generality.** In Appendix A, we show that WAFL is a more general case of adversarial robust FL, and that WAFL can guarantee robustness for all adversarial examples inside the ambiguity set of adversarial robust FL.
>
> In addition to adversarial robustness, we demonstrate that WAFL enjoys properties of domain adaptation. As shown in Section 5 and Figure 1, the flexibility of WAFL with respect to $\rho$ and $\lambda$ means that the nominal distribution $\widehat{P}_{\lambda’}$ can be shifted, leading to an ambiguity set being able to cover a new distribution $Q$. This characteristic is not found in Agnostic FL.
>
>
> > The paper didn't report a systematic way to choose $\gamma$ in the numerical experiment part; the choices of for MNIST and the choices of $\gamma = 0.5$ for MNIST and $\gamma = 10$ for CIFAR-10 are not well-supported.
>
> In Figure 3, $\gamma$ is considered as a hyper-parameter for fine-tuning depending on the heterogeneous and non-i.i.d data and the level of distribution shifts. So the value of $\gamma$ is fine-tuned for different dataset, and different levels  of distribution shifts.
>
> > For the numerical part, it would be better to include (Mohri et al., 2019) or (Deng et al., 2020b) as benchmark as well. This would be helpful to clarify my first point.
>
> Mohri et al., 2019 evaluated their proposed algorithm by comparing the accuracy of the domain agnostic model (model that minimizes $\mathcal{L}_\Lambda$) with the model trained with the union of samples in uniform distribution, and the models trained on individual domains. They did not conduct experiments on domain adaptation or adversarial attacks.
> Deng et al., 2020b proposed DRFA, a distributionally robust FL algorithm based on a minimax formulation, to reduce communication rounds over federated training. They did not mention about the efficiency of their algorithm in dealing with adversarial perturbations, distributional shifts or domain adaptation.
>
> > It is claimed on page 3 that In FL, some existing works have explored the Wasserstein distance to enhance robustness (Reisizadeh et al., 2020; Diamandis et al., 2021; Du et al., 2020; Deng et al., 2020b). Though Du et al., 2020 and Deng et al., 2020b are examples of distributional robust FL, I can not find their connection to Wasserstein distance. Did I miss something here?
>
> Thank you. We would like to clarify that only Reisizadeh et al., 2020 and Diamandis et al., 2021 used the Wasserstein distance in FL. In particular, Reisizadeh et al., 2020 considered the Wasserstein cost to analyze the properties of their proposed minimax formulation, while Diamandis et al., 2021 employed the Wasserstein distance for a mixed linear regression problem in FL, which is different from our approach.
>
> We would like to revise that Du et al., 2020 and Deng et al., 2020b are just the prior works on distributionally robust  FL and, unrelated to the Wasserstein distance.

---

### Official Review · Reviewer_g9vL · 2021-10-29

**Correctness:** 3
**Technical Novelty And Significance:** 1
**Empirical Novelty And Significance:** 2
**Recommendation:** 3
**Confidence:** 4

**Main Review:**

The problem under investigation is important. However, I do not feel this paper made sufficient novel contributions to either the field of WDRO or the field of FL. The duality result (5) used by this paper was not new. The formulation (4) is the canonical form of a WDRO problem, and thus I did not see the difficulty of solving it.

One of my major concerns is in (6), where the authors did not solve \gamma to optimality, but instead considered it as a hyper-parameter. This would break strong duality, meaning that what Algorithm 1 solved is not the original WDRO problem (4), but just a relaxed version, and we do not know the tightness of such a relaxation. Can you explain the rationale of not solving \gamma to optimality?

Algorithm 1 does not seem to be very different from FedAvg, except that the original loss is replaced by the surrogate loss.

Also, at the end of Section 2, the authors mentioned several related works exploring Wasserstein robustness in FL. It would be good to expand a bit on what is the difference between this work and those previous works, what makes your paper novel and what are your contributions. In the experimental section, it would be good to see whether your approach improves upon those past works.

The experiments do not seem to be convincing either. FedAvg itself is not a robust algorithm, and thus I do not think comparing with it gives any insights on the robustness of the proposed approach. The improvement of WAFL starts to be significant only when the proportion of attached clients is very high (~80%), which puts the benefits of using WAFL in doubt.

Minor points: how did you choose \lambda_i? Did you set it to n_i/n?

**Summary Of The Paper:**

This paper presents a Wasserstein distributionally robust optimization (WDRO) framework for federated learning to hedge against statistical heterogeneity. The authors utilized the duality result in previous WDRO works to make their formulation tractable, and proposed a distributed algorithm to solve a relaxed WDRO problem. They conducted several numerical experiments on MNIST and CIFAR-10 to compare their method with FedAvg (non-robust algorithm), FedPGM and FedFGSM (robust algorithms), and demonstrated the advantage of their approach when the proportion of attacked clients is over 80%.

**Summary Of The Review:**

Overall I do not think this paper made sufficient novel contributions to the field of WDRO and FL. Several theoretical results are already well-known by the WDRO community, and the proposed algorithm is not very different from FedAvg. The numerical experiments also do not look very convincing.

---

> ### Author Response · Authors · 2021-11-17
> **Response to Reviewer g9vL (Part 1)**
>
> We would like to thank Reviewer g9vL for their feedback. We take this opportunity to address the reviewer's comments below.
>
>
> > The problem under investigation is important. However, I do not feel this paper made sufficient novel contributions to either the field of WDRO or the field of FL. The duality result (5) used by this paper was not new. The formulation (4) is the canonical form of a WDRO problem, and thus I did not see the difficulty of solving it.
>
> We do not claim that the result (5) is new, and we have cited previous work relating to this duality. Instead, the novelty of our work is the establishment of generalization for FL through the lens of WDRO. We would appreciate it if the reviewer considered further comments on our generalization contributions.
> To the best of our knowledge, the formulation (4) is not amenable to a distributed algorithm design (such as FL algorithms). Having said that, we believe that (4) cannot help us design a distributed algorithm for FL, thus we use the duality in (5) to enable the distributed property of WAFL.
> Since the reviewer claimed that it is not difficult to solve (4) directly, we would appreciate it if the reviewer could give us any pointer to existing work that directly solves Problem (4) in a distributed context (like FL)?
>
>
> > One of my major concerns is in (6), where the authors did not solve $\gamma$ to optimality, but instead considered it as a hyper-parameter. This would break strong duality, meaning that what Algorithm 1 solved is not the original WDRO problem (4), but just a relaxed version, and we do not know the tightness of such a relaxation. Can you explain the rationale of not solving $\gamma$ to optimality?
>
> We agree with the reviewer that this is a relaxation; that is, we treat $\gamma$ as a hyper-parameter and do not aim to solve (5) directly. We also agree that, as a result, strong duality does not hold in this case. The logic behind our choice follows that in (Sinha et al.), which is that by considering $\gamma$ as a hyper-parameter, we relinquish the “prescribed” amount of robustness $\rho$, and focus on the optimization problem (6). As long as $\gamma$ is large enough (corresponding to small enough $\rho$, or “not too much robustness”), the function $\varphi$ (in Appendix C; $\phi$ is the optimal value of $\varphi$) remains easy to optimize.
>
> Regarding the rationale behind considering $\gamma$ as a hyper-parameter, directly solving for $\gamma$ is a computationally costly task, yet if we set $\gamma$ large enough, we ensure an easy inner optimization problem with a moderate level of robustness.
>
>
> > Algorithm 1 does not seem to be very different from FedAvg, except that the original loss is replaced by the surrogate loss.
>
> The reviewer is correct in comparing Algorithm 1 to FedAvg. In fact, if $\gamma$ is set such that the inner problem is easy to solve, WAFL enjoys similar convergence guarantees to those of FedAvg. An attractive property of Algorithm 1 is: by varying $\gamma$, WAFL can provide a different level of robustness.
>
>
> > At the end of Section 2, the authors mentioned several related works exploring Wasserstein robustness in FL. It would be good to expand a bit on what is the difference between this work and those previous works, what makes your paper novel and what are your contributions. In the experimental section, it would be good to see whether your approach improves upon those past works.
>
> We first note that only Reisizadeh et al., 2020 and Diamandis et al., 2021 used the Wasserstein distance in FL. In particular, Reisizadeh et al., 2020 considered the Wasserstein cost to analyze the properties of their proposed minimax formulation, while Diamandis et al., 2021 employed the Wasserstein distance for a mixed linear regression problem in FL, which is different from our approach.
> We would like to revise that Du et al., 2020 and Deng et al., 2020b are just prior works on distributionally robust  FL and unrelated to the Wasserstein distance.
>
> In any robust optimization problem, the ambiguity set is a key ingredient to defining the level of robustness. Therefore, in Section 3, we theoretically analyze Wasserstein robust risk in FL and show that the proposed WAFL is more general than Agnostic FL approach (Deng et al., 2020b) and Adversarially Robust FL approach (Reisizadeh et al., 2020); that is, we show that the ambiguity sets induced by these methods can be inside a Wasserstein ambiguity set with appropriately chosen center and radius.

---

> > ### Author Response · Authors · 2021-11-17
> > **Response to Reviewer g9vL (Part 2)**
> >
> > > The experiments do not seem to be convincing either. FedAvg itself is not a robust algorithm, and thus I do not think comparing with it gives any insights on the robustness of the proposed approach.
> >
> > The main purpose of the comparison with FedAvg is to show the generalization of WAFL. There are 2 scenarios considered in Figure 3 in our manuscript: clean data and distribution shift settings.
> >
> > In the case of clean data, we consider the setting where there is no attacked client, similar to the traditional FL setting. WAFL shows better generalization performance than FedAvg in heterogeneous and non.i.i.d. data settings.
> >
> > In fact, robust methods such as FedFGSM and FedPGM demonstrate worse performance in the clean data settings while FedAvg demonstrates worse performance in the distribution shift setting. By contrast, WAFL not only has good generalization when there is no attack but also when there are attacked clients: for example, in Figure 3 when $\gamma = 0.5$ for MNIST and $\gamma = 10$ for CIFAR-10.
> >
> > > The improvement of WAFL starts to be significant only when the proportion of attached clients is very high (~80%), which puts the benefits of using WAFL in doubt.
> >
> > In Figure 3, we show that WAFL improves over FedAvg even when there is no attacked client. The improvement is significant when the proportion of attacked clients reaches 40% (not only in the case where it is 80%); for example, for MINST, the improvement is 18% compared with FedAvg (Figure 4). However, the benefits of WAFL in terms of robustness are the improvements over FedFGSM, FedPGM, two common robust methods to again data poisoning attack in FL even though both FedFGSM, FedPGM are trained on the same $\epsilon$ attack level.
> >
> >
> > > How did you choose $\lambda_i$? Did you set it to $\frac{n_i}{n}$?
> >
> > The reviewer is correct: In our experiments, we do not try to find the optimal $\lambda$ for domain adaptation. We simply fix $\lambda_i = \frac{n_i}{n}$. This is also the choice of other FL optimization frameworks such as FedAvg.

---

### Official Review · Reviewer_M1m9 · 2021-11-02

**Correctness:** 4
**Technical Novelty And Significance:** 3
**Empirical Novelty And Significance:** 3
**Recommendation:** 6
**Confidence:** 3

**Main Review:**

The paper is well-written and the concepts are explained in technical detail. While the ideas implemented are not surprising, appearing previously in Wasserstein Adversarial learning algorithms, the authors do a decent job of explicating the details of the method. The SGD algorithm via usage of duality properties is standard and the theoretical justification is solid. Overall, I would say this work is a reasonable contribution. I have some comments about the paper.

1. Can you show simple cases of inference where Wasserstein robust optimization does better or specific signal/noise types?


2. Suppose you have $n$ data points and you perform the WAFL method, does it give smaller number of components in terms of the maximizing argument (Q in this case). I feel that invoking Wasserstein robustness should provide a more interpretable classifier. Is that intuition justified? If so, is that confirmable through some experimental validation or otherwise?

3. Please provide the computational time comparison with relevant methods.


**Summary Of The Paper:**

The paper proposes a Wasserstein robust Distributional optimization scheme to provide robust approach for empirical risk minimization. Generalizing the concepts of Agnostic Federated Learning, the method finds applicability in domain adaptation as well as big data settings. The authors also propose and SGD algorithm to solve the problem described and provides a theoretical upper bound for the resultant estimates.

**Summary Of The Review:**

Overall a solid piece of work. Results are not surprising but are reasonable.

---

> ### Author Response · Authors · 2021-11-19
> **Response to Reviewer M1m9**
>
> > Can you show simple cases of inference where Wasserstein robust optimization does better or specific signal/noise types?
>
> With the extreme generality, Wasserstein Distributionally Robust Federated Learning (WAFL) can be effectively applied to many cases of inference in FL such as enhancing robustness to distribution shifts, dealing with adversarial perturbations, providing fairness, and even improving communication efficiency. To address your concerns, we would like to suggest two inference cases that directly benefit from WAFL.
>
> **First, WAFL is highly capable of providing robust inference under distributional shifts.** As we presented in Section 1 of the paper, statistical heterogeneity is one of the critical challenges in FL. Specially, the performance of the global model often degrades significantly on unseen clients’ data distributions. Personalization is a well-known approach to tackle this problem, but it lacks adaptability and scalability. WAFL, on the other hand, relies on the Wasserstein ambiguity set which is rich enough to cover all possible client distributions, so long as the distance between distributions is bounded by a Wasserstein distance. By exploring the natural geometry in the data space, WAFL can achieve good performance in tasks such as unsupervised domain adaptation, domain generalization, etc., while preserving data privacy. In Appendix G.4, we show the performance of the unsupervised domain adaptation task of WAFL on digit datasets, which is overall better than that of the de facto FedAvg.
>
> **Second, WAFL makes FL models more robust against adversarial perturbations.** Instead of rigorously identifying the classes of adversarial attacks to build defense mechanisms, WAFL has a robust generalization bound, guaranteeing small to moderate amounts of robustness. This bound is applicable for all adversarial distributions inside the Wasserstein ball, thus WAFL can be used to model many types and levels of adversarial robustness. In the Experiment section, we demonstrated that WAFL has better performance than other popular robust mechanisms such as projected-gradient method and fast-gradient method (in federated setting, when the proportion of adversary clients is more than 40%). Furthermore, we believe that WAFL can also provide robust inference in tasks such as image transformations, untrusted client data, etc.
>
> > Suppose you have n data points and you perform the WAFL method, does it give smaller number of components in terms of the maximizing argument (Q in this case). I feel that invoking Wasserstein robustness should provide a more interpretable classifier. Is that intuition justified? If so, is that confirmable through some experimental validation or otherwise?
>
> Thank you for the question. As we understand it, the component $Q$ the reviewer mentions is the adversarial distribution $Q$ in problem (4). A Wasserstein distributionally robust method would need to find such a distribution $Q$ in its optimization. Our paper makes use of the results in (5) and (6) to show that problem (4) has a relaxed, but computationally friendlier, formulation, and uses (6) as the optimization problem throughout the rest of the paper. Therefore, we do not explicitly find distribution $Q$.
>
> From the perspective of interpretability, Wasserstein robust optimization first sets up an ambiguity set in which all examples are ensured robustness. While WAFL does not explicitly prescribe the radius of the ambiguity set ($\rho$), it does so implicitly by setting up its dual parameter $\gamma$, based on results in (5). As a result, WAFL guarantees robustness for all examples inside its Wasserstein ambiguity set, and algorithm designers have flexibility in choosing how large this set is and where its center may be.
>
> > Please provide the computational time comparison with relevant methods.
>
> We run WAFL for 200 communication rounds. With the same logistic regression model, DRFA (Deng et al., 2020b) converges in about 100 rounds, and AFL (Mohri et al., 2019) converges in more than 250 rounds.

---

### Decision · Program_Chairs · 2022-01-20

**Decision:**

Reject

**Comment:**

The main motivation of this work is to introduce robustness in federated learning, through a Wasserstein uncertainty set. The end result, however, leaves a mixed feeling: As the reviewers pointed out, the authors, perhaps for computational convenience, forgo strong duality and treat the important variable gamma as a hyperparameter, which renders large part of the work follow immediately from existing work: essentially, we simply use a different loss function in FedAvg. While there may be advantages to choose one loss over another in any specific application, this itself is not a significant contribution. The comparison against existing FL algorithms is also a bit weak: Despite of the reviewer's request, the authors did not compare to other robust FL algorithms (e.g., AFL), thus it is not clear what is the real advantage of the proposed algorithm. As a result, we believe the current draft is not ready for publication.